# Deglacial to postglacial history of Nares Strait, Northwest Greenland: a marine perspective from Kane Basin

Eleanor Georgiadis[1,2], Jacques Giraudeau[1], Philippe Martinez[1], Patrick Lajeunesse[2], Guillaume St-Onge[3], Sabine Schmidt[1], Guillaume Massé[2]

[1]Université de Bordeaux, CNRS, UMR 5805 EPOC, 33615 Pessac, France
[2]Université Laval, UMI 3376 TAKUVIK, Québec, G1V 0A6, Canada
[3]Université du Québec à Rimouski and GEOTOP Research Center, Institut des sciences de la mer de Rimouski (ISMER), Rimouski, G5L 3A1, Canada

*Correspondence to*: Eleanor Georgiadis (eleanor.georgiadis@u-bordeaux.fr)

**Abstract.** A radiocarbon dated marine sediment core retrieved in Kane Basin, central Nares Strait, was analysed to constrain the timing of the postglacial opening of this Arctic gateway and its Holocene evolution. This study is based on a set of sedimentological and geochemical proxies of changing sedimentary processes and sources that provide new insight into the evolution of ice sheet configuration in Nares Strait. Proglacial marine sedimentation at the core site initiated *ca*. 9.0 cal. ka BP following the retreat of grounded ice. Varying contributions of sand and clasts suggest unstable sea ice conditions and glacial activity which subsisted until *ca*. 7.5 cal. ka BP under the combined influence of warm atmospheric temperatures and proglacial cooling induced by the nearby Innuitian (IIS) and Greenland (GIS) ice sheets. An IRD-rich interval is interpreted as the collapse of the ice saddle in Kennedy Channel *ca*. 8.3 cal. ka BP that marks the complete opening of Nares Strait and the initial connection between the Lincoln Sea and northernmost Baffin Bay. Delivery of sediment by icebergs was strengthened between *ca* 8.3 and *ca*. 7.5 cal. ka BP following the collapse of the buttress of glacial ice in Kennedy Channel that triggered the acceleration of GIS and IIS fluxes toward Nares Strait. The destabilisation in glacial ice eventually led to the rapid retreat of the GIS in eastern Kane Basin at about 8.1 cal. ka BP as evidenced by a noticeable change in sediment geochemistry in our core. The gradual decrease of carbonate inputs to Kane Basin between ~8.1 and ~4.1 cal. ka BP reflects the late deglaciation of Washington Land. The shoaling of Kane Basin can be observed in our record by the increased winnowing of lighter particles as the glacio-isostatic rebound brought the seabed closer to subsurface currents. Reduced iceberg delivery from 7.5 to 1.9 cal ka BP inferred by our dataset may be linked to the retreat of the bordering ice sheets on land that decreased their number of marine termini.

# 1 Introduction

The Holocene history of Nares Strait, Northwest Greenland, has remained somewhat cryptic despite investigations during the past four decades (e.g. Blake, 1979; Kelly and Bennike, 1992; Mudie et al., 2004; Jennings et al., 2011.). Nares Strait is a key gateway for Arctic sea water and ice toward the Atlantic Ocean, contributing to up to half of the volume of water transported through the Canadian Arctic Archipelago (CAA) which provides fresh water to the Labrador Sea and influences deep water formation (Belkin et al., 1998; Münchow et al., 2006; McGeehan and Maslowski, 2012). Nares Strait supplies one of the most productive regions of the Arctic, the North Water Polynya (NOW), with nutrient-rich Pacific water (Jones et al., 2003; Jones and Eert, 2004) and maintains its very existence by trapping sea and calved glacial ice in ice arches in the north and the south of the strait (Melling et al., 2001; Mundy and Barber, 2001).

Despite the importance of Nares Strait, intrinsic investigations into its late Pleistocene history, which is intimately linked with the dynamics of the bordering Innuitian (IIS) and Greenland (GIS) ice sheets, are relatively sparse and much of the knowledge relies on land-based studies. Debate initially surrounded early studies into glacial configuration in the CAA with some authors concluding that the CAA channels were not blocked during the Last Glacial Maximum (LGM) (Franklin Ice Complex theory, e.g. England 1976), while others argued that the IIS coalesced with the bordering Greenland and Laurentide Ice Sheets (e.g. Blake, 1970). The presence of erratic boulders originating from Greenland on Ellesmere Island (England, 1999), cosmogenic nuclide surface exposure dating (Zreda et al., 1999) and radiocarbon dating on mollusc shells (e.g. Bennike et al., 1987; Blake et al., 1992; Kelly and Bennike, 1992) finally settled the argument in favour of the latter narrative by supporting the coalescence of the IIS and GIS along Nares Strait between 19 and *ca*. 8 $^{14}$C ka BP (~22-8.2 cal. ka BP, ΔR=240). England (1999) reviewed all land-based evidence available at that time and proposed a complex deglacial history of Nares Strait, featuring the late breakup of glacial ice in central Nares Strait (i.e. Kennedy Channel). These land-based studies have been complemented by Jennings et al. (2011) and Mudie et al. (2004) investigations of marine sediment cores collected in Hall Basin, northernmost Nares Strait which record a change in a number of environmental proxies ca. 8.3 $^{14}$C ka BP (~8.5 cal ka BP, ΔR=240). More recently, the geophysical mapping of submarine glacial landforms by Jakobsson et al. (2018) provided additional insight regarding the retreat of Petermann Glacier in Hall Basin, and new surface exposure dating on moraines in Washington Land demonstrate that the Humboldt Glacier, eastern Kane Basin, abandoned a previous position of stability ca. 8.3 ±1.7 ka BP (Reusche et al., 2018). To date, little is known about the downstream consequences of the opening of the strait, despite the recovery of multiple marine archives in northernmost Baffin Bay (Blake et al., 1996; Levac et al., 2001; Knudsen et al., 2008; St-Onge and St-Onge, 2014). Several aspects of the evolution of northernmost Baffin Bay have been explored with regards to ice sheet retreat in the area (Blake et al., 1996), ice sheet dynamics (St-Onge and St-Onge, 2014) and changes in sea ice conditions and marine productivity during deglacial and postglacial times (Levac et al., 2001; Knudsen et al., 2008; St-Onge and St-Onge, 2014). Unfortunately however, these archives do not cover a continuous record of the Holocene and the sediments deposited around and before the opening of the strait were not recovered or are unable to provide any further information on the timing and consequences of the event.

Here we present sedimentological, geochemical and geochronological data obtained from a 4.25-meter-long marine sediment core (AMD14-Kane2b) retrieved in Kane Basin, central Nares Strait. This core provides a continuous sedimentary record spanning the last 9.0 cal. ka BP, i.e. from the inception of the early Holocene retreat of the GIS and IIS in Nares Strait to modern times. Our set of sedimentological and geochemical records derived from this study presents the first offshore evidence of an ice-free environment in Kane Basin in the Early Holocene and offers a unique opportunity to explore the local dynamics of ice-sheet retreat leading to the opening of the strait and the establishment of the modern oceanographic circulation pattern.

## 2 Regional settings

Nares Strait is a long (530 km) and narrow channel separating Northwest Greenland from Ellesmere Island, Arctic Canada, connecting the Arctic Ocean to the Atlantic Ocean in Baffin Bay (Fig. 1). Kane Basin is the central, wide (120 km large at its broadest point, totalling an area of approximately 27,000 km²) and shallow (220 m deep) basin within Nares Strait. It separates Smith Sound (600 m deep, 50 km wide) in the south of the strait from Kennedy Channel (340 m deep, 30 km wide) in the north. A smaller but deeper basin, Hall Basin (800 m deep), where the Petermann Glacier terminates, connects Kennedy Channel to the Robeson Chanel (400 m deep, 21 km wide) in the northernmost sector of the strait.

The oceanographic circulation in Nares Strait consists of a generally southward flowing current driven by the barotropic gradient between the Lincoln Sea and Baffin Bay (Kliem & Greenberg, 2003; Münchow et al., 2006), while the baroclinic temperature balance generates strong, northerly winds that affect surface layers (Samelson and Barbour., 2006; Münchow et al., 2007; Rabe et al., 2012). The relative influence of the barotropic *vs.* baroclinic factors that control the currents in Nares Strait is highly dependent on the presence of sea ice that inhibits wind stress when landfast (Rabe et al., 2012; Münchow, 2016). Long-term ADCP measurements of flow velocity record average speeds of 20-30 cm.s$^{-1}$ in Kennedy Channel (Rabe et al., 2012; Münchow et al., 2006) and 10-15 cm.s$^{-1}$ in Smith Sound (Melling et al., 2001) with the highest velocities measured in the top 100 m of the water column. Strong currents peaking at 60 cm.s$^{-1}$ have been measured instantaneously in Robeson Channel (Münchow et al., 2007). The speed of the flow decreases in the wider sections of Nares Strait. A northward current has been shown to enter Kane Basin from northern Baffin Bay (Bailey, 1957; Muench, 1971; Melling et al., 2001; Münchow et al., 2007). Temperature and salinity isolines imply that an anti-clockwise circulation takes place in the surface layers of Kane Basin, while the deeper southward flow of Arctic water is channelled by bottom topography and concentrated in the basin's western trough (Muench, 1971; Moynihan, 1972; Münchow et al., 2007).

Sea ice concentration in Nares Strait is usually over 80% from September to June (Barber *et al*., 2001). The state of the ice varies between mobile (July to November) and fast-ice (November to June). The unique morphology of the strait leads to the formation of ice arches in Nares Strait when sea ice becomes landfast in the winter. The ice arches are a salient feature in the local and regional oceanography of Nares Strait: they not only block sea ice from drifting southward in the strait sustaining the existence of the NOW Polynya (Barber et al., 2001), but they also control the export of low salinity Arctic water into Baffin

Bay (Münchow, 2016). The main iceberg sources for the strait are Petermann Glacier in Hall Basin, and Humboldt Glacier in Kane Basin, both outlets of the GIS.

The Greenland coast bordering Kane Basin is relatively flat. In Inglefield Land the Precambrian basement is exposed, displaying supracrustal crystalline rocks and metamorphic rocks, essentially reported as aluminous metasediments and gneisses or granitoid gneisses, with some references to quartzite (Fig. 2, Koch, 1933; Dawes, 1976, 2004; Harrison and Oakley, 2006 and references therein). Dawes (2004) postulated that this Precambrian basement also underlies the 100 km wide Humboldt Glacier, a claim that is supported by the dominance of crystalline material delivered in modern glacimarine sediments in front of the Humboldt Glacier (Fig.2, Kravitz, 1976). To the north, the Precambrian basement in Washington Land is overlaid by Cambrian, Ordovician and Silurian dolomites, limestones and evaporites (Koch, 1929a, b; Harrison et al., 2006 and references therein). The Ellesmere shore of Kane Basin rises abruptly from sea level and is punctured by narrow fjords, penetrating inland for nearly 100 km (Kravitz 1982). In southern Kane Basin, the same Precambrian crystalline rocks outcrop to form the Ellesmere-Inglefield Precambrian Belt. The central and northern sectors of Ellesmere Island's coast mainly comprise Cambrian to Devonian carbonates and evaporates. Fluviodeltaic quartz sandstone, volcanistic sandstone, minor arkose and sometimes coal are found in the Paleogene Eureka Sound sequence that occurs along the western coast of Kane Basin, on the Ellesmere Island flank of Kennedy Channel and on Judge Daly Promontory (Christie, 1964, 1973; Kerr, 1967, 1968; Miall, 1982; Oakey and Damaske, 2004). Coal bearing Paleogene clastics also occur along the coast of Bache Peninsula and in morainic deposits on Johan Peninsula in south-western Kane Basin (Fig. 2, Kalkreuth et al., 1993).

Kravitz (1976) described modern sedimentation in Kane Basin according to three main provinces defined on the basis of mineralogical and grain size characteristics. The first province covers the eastern, central and southern part of the basin in which the predominant crystalline clay and silt sediments are water-transported off Humboldt Glacier and Inglefield Land. The second province, in the west of the basin, includes a higher fraction of ice-transported materials, mostly carbonates with clastic debris occurring in the deeper trough. Northern Kane Basin makes up the third province in which water-transported, mostly carbonate sediments from Washington Land, are deposited in its northernmost part, while ice-transported crystalline particles are more common in the southern part of this province.

## 3 Material and methods

Sediment core AMD14-Kane2b was retrieved at 217 m water depth in Kane Basin, Nares Strait (79°31.140'N; 70°53.287'W) during the 2014 ArcticNet expedition of the CCGS *Amundsen*. This core was collected with a wide-square section (25 cm x 25 cm) gravity corer (Calypso Square or "CASQ") and immediately sub-sampled on-board using large U-channels.

### 3.1 Sedimentological analyses

The description of the various lithofacies was based on the visual description of the core and high-resolution images using a computed tomography (CT) scanner (Siemens SOMATOM Definition AS+ 128 at the Institut national de la Recherche

Scientifique, Quebec, Canada). Changes in sediment density were estimated from variations in the CT-numbers which were processed according to Fortin et al. (2013). To complement CT analyses, a series of thin sections covering two intervals were sampled across major lithological changes toward the base of the core (425-405 cm and 373.5-323.5 cm) in order to visualise the internal structure and examine the nature of these facies. The thin sections were prepared according to Zaragosi et al.

(2006). Grain size analyses were performed at intervals of 2 to 4 cm throughout the archive. A Malvern 2000 laser sizer was used to determine the relative contribution (expressed as % of particles) of clay and colloids (0.04-4 µm), silt (4-63 µm) and sand (63-2000 µm) within the < 2 mm fraction. The same samples were also subjected to wet sieving through 63, 125 and 800 µm meshes in order to determine the weight fraction of sands and identifiable ice-rafted debris (IRD), expressed as % weight of the bulk dry sediment.

**3.2 XRF core-scanning**

High-resolution (0.5 cm) X-Ray Fluorescence (XRF) scanning was conducted along the archive using an AVAATECH XRF core-scanner. The semi-quantitative elemental composition of the sediment was measured throughout the whole archive with the exception of two units which contain large clasts. Measurements were acquired with generator settings of 10, 30 and 50 kV in order to detect elements in the range of Al to Ba. Elemental ratios or normalisation to the sum of all elements except Rh

and Ag, whose counts are biased during data acquisition, were used to minimise the effects of grain size and water content on elemental counts (Weltje and Tjallingii; 2008). XRF core scanner-derived elemental ratios have been used as a time-efficient method to assess down-core variations in grain size (e.g. Guyard et al., 2013; Mulder et al., 2013; Bahr et al., 2014) and/or sediment sources for detrital material in similar high latitude locations (e.g. Møller et al., 2006; Bervid et al., 2017). The applicability of this approach in Kane Basin is tested in the present study by using Ti/K and Fe/Ca as proxies of grain-size and

sediment source, respectively. We also demonstrate a correlation between normalised K counts and clay content in core AMD14-Kane2b.

**3.3 Chronology and radiocarbon dating in Nares Strait**

The chronology is based on a set of 18 radiocarbon ages obtained from mixed benthic foraminifera samples and unidentified mollusc shells. The core top is dated at -5 years BP (1955 AD) based on [210]Pb measurements and a comparison with the [210]Pb

profile (Fig. 3) obtained from a box core collected at the same coring site.

Reservoir ages in Nares Strait are difficult to assess owing to the scarcity of pre-bomb specimens in collections of marine shells from the area. Only three molluscs were dated in Nares Strait with ΔR ranging between ~180 and ~320 years, comparing relatively well with molluscs from the western sector of northernmost Baffin Bay (ΔR of ~140 and ~270), while molluscs collected near Thule, North-western Greenland, yielded negative ΔR (McNeely et al., 2006). Coulthard et al. (2010) proposed

a regional ΔR for the CAA of 335 years based on the McNeely et al. (2006) dataset of pre-bomb radiocarbon dated molluscs and taking into account the general oceanographic circulation in the CAA. However, unlike in other passages of the CAA

which present shallow sills at their southern extremities, younger Atlantic water from Baffin Bay enters Nares Strait – or at least Kane Basin – from the south (Bailey, 1957; Muench, 1971; Münchow et al., 2007). We thus choose to correct [14]C ages in this study with the average ΔR of the three pre-bomb collected mollusc shells in Nares Strait, i.e. 240 ±51 years, bearing in mind that reservoir ages are likely comprise between 0 and 335 years and may have changed through time as a consequence

of the major oceanic reorganisation undergone in Nares Strait. Radiocarbon dating in Nares Strait is further complicated by the proximity of old carbonate rocks that are prone to introducing additional uncertainties in the [14]C ages yielded by deposit-feeding molluscs (England et al., 2013). The non-systematic discrepancies between ages yielded from deposit-feeders and those from suspension-feeding molluscs – the so-called *Portlandia* effect (England et al., 2013) – cannot be corrected. However, this represents a greater challenge for landbound studies that pinpoint the timing of the deglaciation of a given

location based on the oldest mollusc found in that location. In contrast, when establishing the age model of sediment cores, the age *vs*. depth relationship reveals any outliers that can be identified as being either (1) remobilised by ice-rafting, slumping or bioturbation, or (2) potentially affected by the *Portlandia* effect. Hence, we deem the *Portlandia* effect to be of minor concern in the establishment of the age model in this study despite the possible inclusion of deposit-feeders in our radiocarbon dataset. The [14]C ages were calibrated with the Marine13 curve (Reimer et al., 2013) using Calib7.1 (Stuiver et al., 2018) with a marine

reservoir age correction of 640 years (ΔR=240). We computed an age/depth model for core AMD14-Kane2b based on radiocarbon-dated material using CLAM 2.2 (Blaauw, 2010) as a smooth spline with a smoothing level of 0.4 and assuming that a 20 cm long clast-rich deposit (300-320 cm) was deposited near instantaneously at the scale of our chronology.

## 4 Results and interpretations

### 4.1 Age model and sedimentation rates in core AMD14-Kane2b

According to our chronology, core AMD14-Kane2b covers the last 9.0 cal. ka BP (Fig. 4). The comparison of the [210]Pb profiles of core AMD14-Kane2b and the box core collected at the same location reveals the relatively good recovery of the topmost sediments in the CASQ core permitted by the large diameter of this corer (sediment loss of ~5 cm, Fig. 3). Fourteen of the [14]C samples yielded consistent ages along a smooth spline, while four outlying radiocarbon ages were excluded from the age model. Only one mollusc fragment was dated (at 301.5 cm) and yielded an age of >43 ka and is thus clearly remobilised (Table

1). A whole mollusc shell at 238.5 cm yielded a radiocarbon age about 1 ka older than expected and is the only specimen we suspect to be affected by the *Portlandia* effect. Two mixed benthic foraminifera samples yielded ages older than expected and most likely include older specimens.

Major changes in depositional environments, most particularly during the time interval corresponding to the lower half of our sediment core, explain the wide range of sedimentation rates. High sedimentation rates are observed between the base and

~250 cm where they decrease from ~220 cm.ka$^{-1}$ to 30 cm.ka$^{-1}$, and after which sedimentation rates increase to reach 50 cm.ka$^{-1}$ at 120 cm before decreasing again to ~20 cm.ka$^{-1}$ at the top of the core.

## 4.2 Relationship between XRF data, grain size and sediment sources

In modern sediments, the spatial variability of sediment geochemistry in Kane Basin is likely related to their provenance. Heavy crystalline minerals (e.g. garnet and orthopyroxene) occur in the eastern province of the basin in provenance of the Humboldt Glacier and Inglefield Land, whereas carbonates in its western sector are sourced from Ellesmere Island or from Washington Land in its northern sector (Fig. 2, Kravitz 1976). The geochemical composition of modern sediments varies likewise, with, most notably, high concentrations of Fe and Zn in the eastern sector of Kane Basin (Kravitz, 1994). Although the exact chemical variability of the source geological units are not known at present, we consider that sedimentary rocks from eastern Kane Basin and northern Nares Strait are likely rich in Ca, whereas higher concentrations of Fe, Si and K presumably characterize the crystalline rocks of the Ellesmere-Inglefield Precambrian Belt. We propose the use of Fe/Ca in our study to follow the potential erosion of rocks from under the Humboldt Glacier and Inglefield Land (presumably Fe-rich), and from Ellesmere Island (presumably Ca-rich). We then infer the position of the GIS and IIS in relation to the core site and the geological units. It can be noted however that a direct link between the XRF-derived elemental composition of the sediment and the nearby geological units can be compromised by the ubiquitous nature of certain elements in crystalline and sedimentary rocks, along with the sensitivity of elemental signals to grain size when using XRF core scanning. The interpretations of our XRF dataset in terms of sediment sources warrant confirmation by future research into the mineral associations in core AMD14-Kane2b (Caron et al., in prep). The inferred position of the GIS margin in Kane Basin exposed hereafter is however unlikely to be affected by the outcome of the latter study owing to our sedimentological and grain size studies that provide evidence for the distance of the ice margin to the coring site.

XRF counts in core AMD14-Kane2b are largely dominated by Ca and Fe which are anticorrelated. Our records show a good correlation between normalised K counts and clay content in the <2 mm fraction (laser diffraction grain size data) with a correlation factor of $r^2=0.57$ that reaches $r^2=0.73$ by removing 9 outlying data points from the total 150 samples analysed by laser diffraction (Fig. 6, Supplementary Figure S.1). Likewise, there is an excellent correlation between silt content and the Ti/K ratio from the XRF elemental composition data. The correlation factor between % silt and Ti/K is $r^2=0.35$, but rises to $r^2=0.84$ by removing 9 outlying data points (7 of which are different to those removed to improve the correlation between K counts and % clay, mainly from lithological units 3A and 3C presented hereafter). The similar trends of normalised K counts and the Fe/Ca ratio in units 2, 3, 4 and 5 suggest that the clay content and sediment source may be linked, or respond to the same controlling factor.

## 4.3 Lithological units and sedimentological processes

The chirp 3.5 kHz sub-bottom profile obtained prior to core recovery is shown in Figure 5. Given the good recovery of recent sediments at the top of core AMD14-Kane2b (Fig.3), we place the top of core AMD14-Kane2b at the sediment-water interface on this profile. Assuming an acoustic velocity of 1500 m.s$^{-1}$, the base of the core reached a coarse unit (unit 0) shown to continue below the retrieved sediment (Fig. 5), which is likely to have stopped the penetration of the CASQ corer. The high

level of backscatter, discontinuous reflectors and lack of internal coherence in unit 0 are all discriminant acoustic characteristics of diamicton which contains high amounts of unsorted clasts in a clay to silt matrix (Davies et al., 1984). Given the thickness of unit 0, we interpret this diamicton as being either subglacial till or the first glacimarine sediments deposited during the retreat of the marine-based ice sheet margin.

Based on CT-scans and grain size records, five lithological units were defined for core AMD14-Kane2b, each corresponding to specific depositional environments (Fig. 6, Table 2). The sedimentological processes at play will be examined here, while their environmental significance will be considered in the discussion section of this paper.

Unit 1 (425-394 cm, ca. 9.0 cal. ka BP) encompasses three subunits of distinct lithological nature.

Subunit 1A (425-416 cm) consists of high density, occasionally sorted coarse sediment in a clayey matrix, interbedded with
thinner layers of lower density silty clay (Table 2). The base of the coarser laminations show erosional contact with the underlying finer beds (thin sections in Table 2). Grain size analysis reveal large amounts of sand (26-39%) and silt (24-32%) in the <2 mm fraction in this interval. The relative weight of the 125-800 μm and >800 μm fractions also contribute considerably to the overall weight of the sediment (18% and 11%, respectively).

These laminated deposits display all the characteristic of ice proximal deposits (List, 1982; Ó Cofaigh and Dowdeswell, 2001).
Unit 1A was most likely deposited at the ice sheet margin some ~9.0 cal. ka ago, according to a dated mollusc shell at the base of unit 1C and given the very high sedimentation rates in ice-proximal environments.

Subunit 1B (416-410 cm) displays a sharp decrease in sediment density with the replacement of sand by finer material (60% clay in the <2 mm fraction, Fig. 6, Table 2). While subunit 1B encompasses some clasts (CT-scan in Table 2), the amount of sand and silt actually present in 1B may be lower than reflected by the laser diffraction and wet-sieving data, as the analysed
samples likely included coarser material from the over- and underlying subunits 1A and 1C. XRF data for subunit 1B show high Fe/Ca and Ti/Ca ratios, and low Ca counts.

The finer grain size in this subunit is indicative of a change from an ice margin to an ice-proximal glacimarine environment where suspended matter settling from turbid meltwater plumes is likely the main depositional process (Elverhøi et al., 1980; Syvitstki, 1991, Dowdeswell et al., 1998; Hogan et al., 2016), although the limited thickness (4 cm) of subunit 1B is rather
unusual for this process. The geochemical grain size tracers Ti/K and K show poor correlation with the relatively low silt and high clay content in subunit B2. While K counts are low, Ti/K ratios are high which may suggest a high energy environment, supporting the previous hypothesis of an ice marginal environment where meltwater pulses can transport relatively large particles. High Fe/Ca are evocative of an eastern origin of the sediments in subunit 1B, implying that the GIS was close to the core site.

Subunit 1C (410-394 cm) interrupts the fine grained sedimentation with a sharp increase in the occurrence of outsized clasts. The coarser fractions account for a significant part of the sediment (up to 18% for both the >800 and 800-125 μm fractions) within a dominantly clayey matrix. Sediment density in subunit 1C increases to reach values similar to those observed in subunit 1A. However, unlike subunit 1A, subunit 1C is not laminated and clasts are larger (frequent gravel) and ungraded.

Given the high gravel content in subunit 1C, we consider that the clasts were predominantly iceberg-rafted to the core location rather than sea-ice rafted (Pfirman et al., 1989; Nürnberg et al; 1994). These large amounts of IRD among very poorly sorted material can be interpreted as (1) increased iceberg calving rates, (2) changes in the delivery of sediment by icebergs (increased melting of or dumping from icebergs) or (3) a severe decrease in the delivery of finer particles that increases the apparent
contribution of clasts to the sediment (Hogan et al., 2016 and references therein).

Unit 2 (394-320 cm, 9.0-8.3 cal. ka BP) can be divided into two subunits based on grain size and density. The relative weight of the coarse fraction varies throughout unit 2 with a generally decreasing trend.

Subunit 2A (394-370 cm, 9.0-8.8 cal. ka BP) is composed of poorly sorted, bioturbated sediment (~55% clay and ~38% silt in the <2 mm fraction) with varying contributions of coarser material (between ~0 and 5%) and occasional lonestones (Fig. 6,
Table 2). Sediment density is fairly high, but gradually decreases toward the top of subunit 2A. Ti/K decreases gradually in this subunit, mirroring the decrease in density and opposing the increase in K counts. The Fe/Ca ratio is low at the base of subunit 2A before increasing upward in this subunit.

The dominance of fine particles in subunit 2A with occasional clasts points to a delivery by meltwater plumes and iceberg-rafting. The decreasing Ti/K and silt content along with increasing K counts and clay suggests a growing distance of the ice
margin from the core site since coarser silts and Ti-bearing minerals settle closer to the ice margin, while clay particles tend to sink in more ice-distal locations (Dowdeswell et al., 1998; Ó Cofaigh and Dowdeswell, 2001). Increasing Fe/Ca in subunit 2A may indicate a growing contribution of Paleozoic carbonates on Ellesmere Island in western Kane Basin and/or Washington Land in northern Kane Basin (Fig. 2).

The sediments of subunit 2B (370-320 cm, 8.9-8.3 cal. ka BP) have a lower density and a lower sand and silt content than
those of subunit 2A, while clay content reaches maximum values to an average 63%. Scarce lonestones occur in this subunit and the sediment appears to be faintly laminated. Four biogenic carbonate samples, both mollusc and mixed benthic foraminifera samples, were dated in subunit 2B and high sedimentation rates of ~130 cm.ka[-1] decreasing upward to 90 cm.ka[-1] were calculated from the age model (Table 1, Fig. 4). Subunit B2 is characterised by low Ti/K and high K and Fe/Ca.

These high sedimentation rates, substantial concentrations of clay and the slightly laminated aspect of subunit 2B indicate that
these sediments were mainly delivered by meltwater plumes in a more distal glacial setting (Ó Cofaigh and Dowdeswell, 2001). Relatively high Fe/Ca possibly reflect an increased contribution from eastern Kane Basin gneisses.

Unit 3 (320-300 cm, 8.3 cal. ka BP) stands out as a clast-rich interval. The high density of this unit is comparable to that of subunits 1A and 1C. CT-scans and thin sections reveal the presence of a finer grained horizon enclosed between coarser material, dividing this interval into three subunits (Table 2).
Subunit 3A (320-313 cm) corresponds to the lower clast-rich subunit. A significant portion of the bulk sediment is attributed to 800-125 μm sand (17% wt) and >800 μm sand (up to 7% wt), while the clay matrix contributes to ~53% of the <2 mm fraction. Ti/K ratios are high whereas Fe/Ca ratios and K have significantly decreased compared to the underlying subunit 2B (Fig. 6).

The high clast content and absence of grading suggest that the sediments forming subunit 3A were ice-rafted and deposited at the core location (Ó Cofaigh and Dowdeswell, 2001). The predominant carbonate (low Fe/Ca) material in this subunit likely originates from northern and/or western Kane Basin.

Faint laminations are visible on the CT-scan images of subunit 3B (313-305 cm). The sediment of this subunit is composed
essentially of clay and silt (47 and 43% respectively) with a relatively low sand content (<10 % in the <2 mm fraction and each of the coarser fraction represents less than 3% of the sediment weight). Ti/K ratios have slightly decreased relatively to 3A, but remain high and display a slightly increasing trend. K counts and Fe/Ca ratios remain low. Analysis of the sieved residues revealed the presence of benthic foraminifera in this subunit which were picked and dated at ~9.4 [14]C BP (9.9 cal. ka BP with ΔR=240, Table 1).

The poor sorting of sediments in subunit 3B could possibly indicate that they were ice-transported, but the near absence of clasts (e.g. in contrast to the overlaying and underlying subunits of interval 3) contradicts this hypothesis. The modest contribution of clay along with the relatively high silt content rather points to the transport and deposition of these sediments by a high velocity current. The elemental signature of this subunit (low Fe/Ca) denotes a probable northern and/or western Kane Basin origin. Concerning the old age yielded from the mixed benthic foraminifera picked in this subunit, the age model
shows that these foraminifera were remobilised. It is possible that a small quantity of pre-Holocene foraminifera was mixed in with living fauna. This would imply that sediments pre-dating the last glaciation (>22 cal. ka BP) were preserved under the extended GIS and IIS in Nares Strait, and were eroded and transported to the core site during the deposition of subunit 3B. An alternative explanation is that the sample is composed of postglacial specimens of a similar age which were eroded from the seabed and transported to the site.

Subunit 3C (305-300 cm) contains large amounts of coarse material with an average of 44 % sand and only 32 % clay in the <2 mm fraction. The sand in this subunit is coarser than in 3A with the 800-125 µm fraction contributing to ~34% of the total sediment while up to a further 10% of the sediment weight is accounted for by the >800 µm fraction. Ti/K ratios (high) and K counts (low) are similar to subunit 3A, whereas Fe/Ca ratios are high in subunit 3C (Fig.5).

The very high clast content of subunit 3C along with high Ti/K ratios and the absence of grading are indicative of iceberg-
rafting and deposition. The shell fragment that was dated in the topmost horizon of this subunit (>42 [14]C BP) was clearly remobilised, likely by ice-rafting. The sediment forming subunit 3C appear to originate from eastern Kane Basin (Fig. 2), given the high Fe/Ca ratio. The age model points to rapid sedimentation of unit 3 with an age of 8.22 cal. ka BP on mixed benthic foraminifera picked from the horizon directly above unit 3, and an age of 8.38 cal. ka BP in a sample 7 cm below the base of unit 3 that extrapolates to ~8.29 cal. ka BP at 320 cm in the age model (Table 1, Fig. 4).

Unit 4 (300-280 cm, 8.3-8.1 cal. ka BP) has a similar density and clay content (~58 %) to subunit 2B. The contribution of sand in these sediments is however higher than in subunit 2B with ~6.5 % weight accounted for by >125 µm sand and ~14 % sand in the <2 mm fraction. The elemental composition of unit 4 is also fairly similar to that of subunit 2B. Ti/K ratios are low while K counts and Fe/Ca ratios are high(Fig. 6).

The high clay content of unit 4 suggests that delivery from meltwater plumes was the dominant sedimentary process at play during this time interval. The substantial amount of sand in this unit indicates that a significant proportion of the sediment was also ice-rafted to the location. As previously mentioned, the increase in ice-rafted debris can indicate (1) increased calving rates when originating from iceberg-rafting, (2) changes in iceberg delivery of sediment (increased melting or dumping of icebergs) or (3) a decrease in the delivery of finer particles that increases the apparent contribution of clasts to the sediment (Hogan et al., 2016 and references therein). The high sedimentation rates (~90 cm.ka$^{-1}$ from the age model and ~190 cm.ka$^{-1}$ from the linear interpolation between the dates at 297.5 cm (8.22 cal. ka BP) and 273.5 cm (8.09 cal. ka BP), Table 1, Fig. 4) support this narrative of delivery by meltwater and ice-rafting that are typically responsible for the transport and deposition of large quantities of sediment (Svendsen et al., 1992; Dowdeswell et al., 1998), while seemingly excluding the possibility of a significant decrease in the delivery of finer particles. High Fe/Ca values suggest that a notable portion of the sediments originates from the Precambrian gneisses of eastern Kane Basin while the slightly decreasing trend displayed by this elemental ratio could potentially be linked to a progressive increase in the contribution of carbonate-rich formations from northern and/or western Kane Basin in this interval.

Unit 5 (280-0 cm, 8.1-0 cal. ka BP) clearly differs from underlying units with regard to the <2 mm grain size fraction (Fig. 6). The clay content drops to steady, lower values (49% on average) and the CT-scans show a generally homogenous sediment with frequent traces of bioturbation. Changes in grain size divide unit 5 into two subunits.

The sediments in subunit 5A (280-250 cm, 8.1-7.5 cal. ka BP) contain a relatively high proportion of sand peaking at 12 % in the <2 mm fraction, while the combined contribution of the coarser fractions averages at ~5.5 % weight. Lonestones occur frequently and are visible in the CT-scan images. K counts and Fe/Ca ratios drop sharply to lower values at the base of subunit 5A (Fig. 6). Ti/K is low, but increases very discreetly toward the top of this subunit.

The significant decrease in clay particles in subunit 5A compared to units 4 and 2B suggests that delivery from meltwater plumes was reduced in this interval, either in relation to a decrease in glacial melting rates or to a more ice-distal setting. The scarcity of clasts in this subunit can be explained by a decrease in marine termini of the GIS and/or IIS, a change in the sea-ice regime and/or the counterparts of the aforementioned hypothesis presented in Hogan et al., 2016, i.e. (1) decreased iceberg calving rates or (2) decrease iceberg melting. We rule out hypothesis 3 (i.e. increased contribution of finer particles) given the reduced contribution of the finer particles in the <2 mm fraction and the decrease in sedimentation rates from 60 to 40 cm.ka$^{-1}$ in this subunit. The sharp decrease in the Fe/Ca ratio between unit 4 and unit 5 is interpreted as a sudden reduction in the contribution of gneissic material in the sediments of core AMD14-Kane2b.

Subunit 5B (250-0 cm, 7.5-0 cal. ka BP) is generally homogenous with lonestones occurring sporadically throughout. The silt content increases gradually from ~40 to ~47 % toward the top of the core. The contribution of the coarser fractions to the total sediment weight is fairly stable from the base to ~40 cm (1.9 cal. ka BP), where the 63-125 μm and >125 μm fractions account for ~2% and <1% of the total sediment weight, respectively. The relative weight of the 63-125 μm sand fraction doubles to ~4% in the top 40 cm of the core (Fig. 5). Both Fe/Ca and Ti/K ratios increase gradually until ~120 cm (~4.1 cal. ka BP) after which they remain relatively high until the core top.

A sample of mixed benthic foraminifera yielded a radiocarbon age some 2 ka older than expected at 238.5 cm. This sample probably contains a mixture of coeval and remobilised foraminifera (either by bioturbation or by water/ice transport from another location).

The overall limited contribution of the coarser fractions to the sediment of subunit 5B in comparison to the underlying lithologic units indicates that ice-delivery of sediment was reduced during this interval. Furthermore, the relatively low amounts of clay imply that meltwater delivery was also weakened. The sediments of subunit 5B were likely primarily water-transported to the core site (Hein and Syvitski, 1992; Gilbert, 1983). The increase in silt and Ti/K toward the top of the core suggest winnowing by an increase in bottom current (Mulder et al., 2013; Bahr et al., 2014). Relatively low sedimentation rates (20-50 cm.ka$^{-1}$) corroborate the narrative that delivery from meltwater plumes was limited in favour of a more hemipelagic sedimentation regime, also supported by the visible bioturbation in this subunit. The increase in fine sand in the most recent sediment may be due to a resumption of ice-rafting over the last 1.9 cal. ka BP. The gradually increasing trend of Fe/Ca suggests that the contribution of carbonates from northern and/or western Kane Basin diminishes gradually between ~270 cm (~7.9 cal. ka BP) in subunit 5A and ~120 cm (~4.1 cal. ka BP) in subunit 5B after which it remains stable until the top of the core.

## 5 Discussion

Our study of core AMD14-Kane2b has enabled us to reconstruct a succession of depositional environments in Kane Basin following the retreat of the formerly coalescent GIS and IIS in Nares Strait (Fig. 8). Here we discuss our reconstructions in the light of other paleoceanographic and paleoclimatic studies to provide a broader view of the Holocene history of Nares Strait (Fig. 6, 7 and 8, Table 2).

Previous studies have shown that the presence of erratic Greenland boulders on Ellesmere Island from Kennedy Channel to the northern entrance of Nares Strait attest to the coalescence of the IIS and GIS along the western side of northern Nares Strait during the Last Glacial Maximum (LGM) (e.g. England, 1999). The absence of such erratics along the western and southern coasts of Kane Basin implies that the confluence of the two ice sheets laid further at sea in the southern half of the strait (England, 1999). Radiocarbon dating on samples from raised beaches provides minimum ages for marine ingress in Nares Strait. These ages are older in the northern and southern extremities of the strait, while only younger ages are yielded by samples in northern Kane Basin and Kennedy Channel implying that a central (grounded) ice saddle persevered longer in the shallower sector of the strait (England, 1999 and references therein; Bennike, 2002). In addition to providing minimum ages for ice sheet retreat, $^{14}$C dating on marine derived material in raised beaches enables one to identify the former shoreline and assess the glacio-isostatic readjustment of the continental crust. However, this approach can only provide minimum ages for (glacial ice-free) aquatic environments at a given place and time and does not necessarily correspond to the position of the ice margin which can be several kilometres inland. Cosmogenic nuclide surface exposure dating is an efficient method to temporally constrain inland ice sheet retreat. However, such investigations are scarce in Nares Strait: only two study

documenting the glacial retreat on Hans Island, off Greenland in Kennedy Channel (Zreda et al., 1999), and in Washington Land (Reusche et al., 2018). England's (1999) paleogeographical maps of ice sheet retreat in Nares Strait based on radiocarbon dated molluscs were revised in Fig. 8 where the offshore limits for the GIS and IIS are proposed based on our sedimentological and geochemical data from core AMD14-Kane2b. The continuous nature of our record also allows us to propose a more precise chronology of the deglaciation of central Nares Strait.

### 5.1 *Ca*. 9.0 cal ka BP: ice sheet retreat in Kane Basin

Our archive demonstrates that marine sedimentation took place in Kane Basin as early as ~9.0 cal. ka BP. Grain size characteristics and sedimentary structures suggest that the laminated basal unit (1A) represents the topmost deposits in the ice marginal environment shortly after ice sheet retreat at the core site (Fig. 5 and 8b, Table 2). The settling of meltwater plume sediments in the proximal glacial marine environment that followed (1B) is devoid of IRD and seems to have been interrupted by an iceberg-rafted interval (1C). The absence of molluscs pre-dating 8.8 cal. ka BP in Kane Basin (England, 1999) likely indicates that following the deglaciation of Smith Sound *ca*. 9.9 cal. ka BP (Fig.8a-c, England, 1999), ice sheet retreat in Kane Basin occurred off the current coast where melting was potentially enhanced by the increasing influence of warmer Atlantic water from the West Greenland Current after 10.9 cal. ka BP, $\Delta R= 0$ (Funder 1990; Kelly et al., 1999; Knudsen et al., 2008). Based on the sedimentary properties of subunit 1A, we propose that *ca*. 9.0 cal ka BP, the GIS/IIS ice margin was located at the core site, completing the offshore area of England's (1999) paleogeographical map for this period (Fig. 8b). The IRD-rich unit 1C which appears to have been deposited by intense ice calving could potentially mark the opening of Kennedy Channel. Our age of ~9.0 cal. ka BP for this unit agrees relatively well with the inferred age of an IRD-rich unit in a sediment core from Hall Basin that was interpreted as the opening of Kennedy Channel at ~8.6 cal. ka BP ($\Delta R=240$) (Jennings et al., 2011). Alternatively, unit 1C could have been deposited during a readvance of the IIS/GIS in Kane Basin in relation to a cold event. Laurentide Ice Sheet readvances have been documented through the dating of end and lateral moraines on Baffin Island aged between 9 and 8 cal. ka BP (Andrews and Ives, 1978) and have been linked to colder periods. A particularly cold event *ca*. 9.2-9.3 cal. ka BP, which is reported in the regional literature from ice core (Vinther et al., 2006, Fisher et al., 2011) and lacustrine records (Axford et al., 2009), may be the source of the calving event in Kane Basin that led to the deposition of unit 1C. Reservoir ages in Kane Basin are likely to have been reduced prior to the collapse of the IIS/GIS ice saddle in Kennedy Channel and the arrival of poorly ventilated Arctic water. The age of unit 1 with $\Delta R=0$ is 9.3 cal. ka BP which suggests to us that subunit 1C could well have been deposited during the 9.2-9.3 cal. ka BP cold event.

### 5.2 9.0-8.3 cal. ka BP: ice proximal to ice distal environment in Kane Basin

The increasingly finer particles that compose unit 2 suggest a growing distance between the core site and the ice margin. The dominant sedimentary process at play is settling from meltwater plumes which is typically responsible for high sedimentation rates, along with frequent delivery of IRD (Table 2). The Early Holocene was characterised by high atmospheric temperatures

during the Holocene Thermal Maximum (HTM) occasioned by greater solar insolation (Bradley, 1990). The HTM has been defined for the eastern sector of the CAA as the period between 10.7 and 7.8 cal. ka BP based on the Agassiz ice core record (Lecavalier et al., 2017). The high melting rates of the ice sheets during the HTM (Fisher et al., 2011) likely enhanced the delivery of particles by meltwater and contributed to the high sedimentation rates observed in our core. More distant glacial

ice from the site is also in good agreement with the occurrence of molluscs dated between 8.8 and 8.4 cal. ka BP on Ellesmere Island and northwest Greenland (Fig. 8c, England et al., 1999). The elemental signature of subunit 2B may suggest however that the GIS was still present in eastern Kane Basin and delivered material derived from the gneiss basement to the core site. The volcanic clastics on Ellesmere Island may also have contributed to Fe counts in our geochemical record, but we consider their input marginal given the limited surface of this geological unit compared to the gneiss and crystalline basement which

outcrops in much of Inglefield Land and underlays Humboldt Glacier (Dawes, 2004). Furthermore, the IIS was a cold base ice sheet (e.g. Tushingham, 1990, Dyke, 2002) and as such likely delivered overall less sediment from meltwater than the warm-based GIS. The occurrence of IRD in unit 2 may imply that relatively open water conditions occurred during this interval, enabling icebergs to drift in Kane Basin, although this may simply be a consequence of high calving rates as the GIS and IIS retreat. Reduced sea ice occurrence in Kane Basin during the Early Holocene would be in good agreement with low sea ice

concentrations reported nearby in Lancaster Sound (from 10 to 6 cal. ka BP, $\Delta R=290$ Vare et al., 2009; from ~10-7.8 cal. ka BP, R=335 Pienkowski et al., 2012). However, while the decreasing trend of the coarse fraction in unit 2 may indicate more stable sea ice conditions (or decreasing calving rates) toward the end of the interval, fluctuations in the coarse fractions in our record may also suggest that sea ice conditions were variable. This is in line with both decreasing atmospheric temperatures towards the end of the HTM (Lecavalier et al., 2017) and Knudsen's et al. (2008) observations of variable West Greenland

Current influence and sea ice conditions between 9.5 and 8.2 cal. ka BP in northernmost Baffin Bay.

**5.3 8.3 cal. ka BP: the opening of Kennedy Channel**

Unit 3 appears to be primarily iceberg-rafted, with an inclusion of a finer, water-transported silty subunit (3B). A foraminifera-derived radiocarbon age obtained from subunit 3B (Table 1) suggests sediment remobilisation within this time interval. If we consider that unit 3A was deposited by the passing over Kane Basin of glacial ice having broken-up in Kennedy Channel, then

a plausible origin for unit 3B could be the entrainment of sediment from northern Nares Strait associated with the discharge of large amounts of water as the connection was established. The absence of any molluscs in Kennedy Channel pre-dating 8.1 cal. ka BP further suggests that Kennedy Channel was still blocked until then, although this method can only provide minimum ages for ice sheet retreat (Fig 8e-g). This proposed age for the opening of Kennedy Channel is only slightly younger than that proposed by Jennings et al. (2011), i.e. ~8.6 cal. ka BP ($\Delta R=240$, Fig.7) based on the estimated age of an IRD event in Hall

Basin, northern Nares Strait (core HLY03-05GC). Both ages can be reconciled assuming that bottom waters in Hall Basin were probably poorly ventilated before the opening of the strait inducing a higher reservoir age in the northern sector of Nares Strait. Furthermore, one might consider that the transitional IRD-rich unit in core HLY03-05GC that is interpreted by Jennings et al. (2011) as representing the opening of Kennedy Channel might in fact represent instabilities in the GIS/IIS prior to–– and

eventually leading to – the complete opening of the strait. If so, the transition from laminated to bioturbated mud in the Hall Basin sediment record which, according to X-radiography, CT-scans and the age model of core HLY03-05GC, occurred close to 8.5 cal. ka BP, ie. ca. 100 years after the deposition of the IRD rich-unit (Jennings et al. 2011), might in fact represent the true opening of Nares Strait (i.e. change from a rather confined Hall basin to a ventilated environment under the influence of a strong southward current). Finally, we assume that the collapse of glacial ice in Kennedy Channel was more likely to have been recorded as an IRD-rich interval south of the channel (i.e. Kane Basin) in the direction of the presumable southward flow, rather than to the north.

It has recently been demonstrated that the Humboldt Glacier retreated from a previous position of stability *ca.* 8.3 ±1.7 ka BP based on surface exposure dating of an abandoned lateral moraine in Washington Land (Reusche et al., 2018). This instability in the Humboldt Glacier may have been linked to the break-up of glacial ice in Kennedy Channel. Furthermore, the onset of decreasing landfast sea ice on the northern coast of Ellesmere Island and northern Greenland after 8.2 cal. ka BP (England et al., 2008; Funder et al., 2011) may have been associated with the flushing of ice through Nares Strait after the opening of Kennedy Channel. The local temperature drop recorded in the Agassiz Ice Core (Lecavalier et al., 2017) and, as suggested by Reusche et al. (2018), in Baffin Bay lacustrine records (Axford et al., 2009), may have been associated with oceanographic and atmospheric reorganisation resulting from the opening of Kennedy Channel, as well as the "8.2 event". Given the excellent correspondence between the aforementioned evidence, we consider that subunits 3A and 3B were deposited as the ice saddle in Kennedy Channel broke-up. The high carbonate signal in the elemental data (Fig. 6) also suggests that the sediments from subunits 3A and 3B originated from northern Nares Strait (Fig. 2) rather than the Humboldt Glacier alone.

The dominant depositional process in subunit 3C is iceberg-rafting based on the abundance of clasts in this interval. The elemental composition of subunit 3C suggests that the sedimentary material likely originates from the GIS in eastern Kane Basin (Fig. 2). Investigations into the internal stratigraphy of the GIS and their comparison to north Greenland ice cores have demonstrated that the collapse of the ice saddle in Kennedy Channel triggered the acceleration of glacial fluxes along Nares Strait (MacGregor et al., 2016). The destabilisation of the GIS following the collapse of the ice saddle may have provoked intense calving that led to the deposition of subunit 3C. In this regard, intense calving of the Humboldt Glacier as recently dated by Reusche et al. (2018) at 8.3 ±1.7 ka BP might explain the observed elemental signature of the top part of unit 3. However, Reusche's et al. (2018) findings also offer an alternative scenario for the deposition of unit 3. Intense calving of the Humboldt Glacier may have occurred as it retreated in eastern Kane Basin and abandoned a lateral moraine in Washington Land *ca.* 8.3 ±1.7 ka BP (Reusche et al., 2018). This alternative scenario alludes to the possibility that the opening of Kennedy Channel may rather have occurred ca. 9.0 cal. ka BP (unit 1C). The elemental signature of subunit 3A and 3B does not, however, point to an eastern source and rather supports a northern/western origin of these sediments.

### 5.4 8.3 - 8.1 cal. ka BP: Increase iceberg delivery to Kane Basin

The abundance of iceberg-rafted debris has increased considerably in unit 4 compared to unit 2. This is possibly the result of the aforementioned acceleration of the GIS and IIS along Nares Strait following the collapse of the ice saddle in Kennedy

Channel (MacGregor et al., 2016), as well as the arrival of icebergs from new sources to Kane Basin situated in northern Nares Strait. The retreating GIS in eastern Kane Basin was likely a primary source of icebergs during this period. However, the high clay content in our record implies that the GIS was still relatively close to the core site and had not yet fully retreated in eastern Kane Basin, contributing to the high sedimentation rates recorded in this unit (Fig. 8e).

**5.5 8.1-7.5 cal. ka BP: Rapid retreat of the GIS in Kane Basin**

The abrupt decrease in clay content and sedimentation rates at 280 cm in our record imply that the ice margin abruptly retreated *ca*. 8.1 cal. ka BP (Fig.6, Fig. 8f). The equal drop in Fe/Ca ratios suggests that it was probably the GIS that retreated rapidly in eastern Kane Basin. This abrupt retreat may have been initiated by the removal of the glacial buttress in Kennedy Channel (unit 3). The subsequent decrease in the >125 µm fraction may be associated with the onset of the deceleration of glacial fluxes along Nares Strait, as well as more distant glacial ice in eastern Kane Basin resulting from the retreat of the GIS. The timing of the retreat of the GIS in eastern Kane Basin corresponds remarkably well with the aforementioned abandonment of a lateral moraine by the Humboldt Glacier (Reusche et al., 2018). The authors in this recent study warn that two samples may be contaminated by previous exposure and that the age of abandonment of the moraine is likely to be younger than the proposed 8.3 ±1.7 ka BP. Given the uncertainties in our radiocarbon dataset (analytical errors and ΔR uncertainties) and in Reusche et al.'s (2018) surface exposure dataset, both our dating of the opening of Kennedy Channel and the retreat of Humboldt Glacier are within the uncertainty rage of the dating of the abandonment of the moraine by the Humboldt Glacier. It is thus difficult to distinguish whether this event was linked to the deglaciation of Kennedy Channel at ~8.3 cal. ka BP, or whether it was delayed until ~8.1 cal. ka BP, after the cold "8.2 event" that may have brought a short period of stability to the GIS.

**5.6 7.5-0 cal. ka BP: deglaciation of Washington Land**

The low Fe/Ca at the beginning of this interval are likely related to the erosion and delivery of material from Washington Land and a decrease in the delivery of crystalline material by the GIS (Fig. 6). The progressive increase in Fe/Ca between 7.5 and 4.1 cal. ka BP can be linked to the deglaciation of Washington Land. The oldest molluscs found on the southern coast of Washington Land are dated between 7.8 and 7.5 cal. ka BP, while specimens found in morainic deposits imply that the extent of the GIS reached a minimum between 4 and 0.7 cal. ka BP (Fig. 7, Bennike, 2002). The decrease of the coarser fractions in our core after ca. 7.5 cal. ka BP may be the result of reduced marine termini of the GIS and hence less calving, as the Greenland coast became deglaciated (Fig. 8g, Bennike, 2002). Increasing silt and Ti/K in our core suggest winnowing by stronger bottom water currents. We propose that as the glacio-isostatic rebound lifted the continental crust in Nares Strait, the seabed was progressively brought closer to the stronger subsurface currents. The isostatic rebound in Kane Basin has been estimated to be between 80 and 120 m (England, 1999 and references therein) which would have had considerable consequences on bottom water velocities. Interestingly, increased sedimentation rates in Kane Basin between ~4.5 and 2.8 cal. ka BP (Fig.3) coincide with a period of atmospheric warming recorded in the Agassiz ice core (Lecavalier et al., 2017). These higher sedimentation rates may have been associated with increased delivery of sediment by meltwater from the GIS and the residual ice caps on

Ellesmere Island during a warmer period. The increase in the contribution of the coarse fraction in core AMD14-Kane2B over the last 1.9 cal. ka BP is suggestive of minimal seasonal sea ice and/or higher calving rates over the last two millennia in Kane Basin. This broadly coincides with low absolute diatom abundances in northernmost Baffin Bay, attesting to poor productivity rates after 2.0 cal ka BP, ΔR=0 (Knudsen et al., 2008). The "bridge dipole" between Kane Basin and northernmost Baffin Bay

entails that when sea-ice conditions in Kane Basin are strong, surface conditions to the south of Smith Sound are largely open and the NOW Polynya is productive, and *vice versa* (Barber et al., 2001). This inverse relationship between sea-ice conditions in Kane Basin and northernmost Baffin Bay has probably been true for at least the past *ca*. 2 cal. ka BP. Recent instabilities in the ice arch in Kane Basin that have led to increased sea-ice export towards northernmost Baffin Bay have been observed by satellite imagery and hence are only documented for the past few decades. Together with Knudsen's et al. (2008) study in

northern Baffin Bay, our results suggest that these instabilities may have begun as early as *ca*. 2 cal. ka BP. Late Holocene decreases in sea-ice occurrence, indicative of milder conditions, were also documented in other sectors of the CAA such as in Barrow Strait between 2.0 and 1.5 cal. ka BP (Pienkowski et al., 2012) or in the adjacent Lancaster Sound between 1.2 and 0.8 cal. ka BP (Vare et al., 2009).

## 6 Conclusion

Our investigation of core AMD14-Kane2b has provided, for the first time, a paleo-environmental reconstruction in Kane Basin over the last *ca*. 9.0 cal. ka. The confrontation of our dataset with both land-based (England, 1999; Bennike, 2002, Reusche et al., 2018) and marine (Jennings et al., 2011) evidence offers several alternative paleo-environmental interpretations for our record. Of particular interest is the determination of which of the two IRD-rich units (unit 1c ~9.0 cal. ka BP, or unit 3 ~8.3 cal. ka BP) in core AMD14-Kane2b might represent the opening of Kennedy Channel. We consider that the evidence is in

favour of a later collapse of glacial ice in Kennedy Channel *ca*. 8.3 cal. ka BP that may have been linked to instabilities in the Humboldt Glacier *ca*. 8.3-8.1 cal. ka BP (Reusche et al., 2018). Our findings concerning the successive paleo-environments in this central sector of Nares Strait following ice sheet retreat can be summarised as followed.

While evolving from a short-lived ice-proximal depositional environment at ~9.0 cal. ka BP to a rather secluded and narrow bay as the ice sheets retreated, compelling evidence indicates that Kane Basin was not connected to Hall Basin until the collapse

of the GIS/IIS saddle in Kennedy Channel at ~8.3 cal. ka BP. The collapse of the glacial buttress in Kennedy Channel may have triggered the acceleration of glacial fluxes toward Nares Strait, increasing calving and iceberg-rafted debris in Kane Basin between 8.3 and 7.5 cal. ka BP. Instabilities in the GIS eventually resulted in the rapid retreat of glacial ice from eastern Kane Basin at 8.1 cal. ka BP. As the basin underwent shoaling induced by the glacio-isostatic rebound, the retreat of the GIS in Washington Land gradually reduced inputs of carbonate materiel to Kane Basin. A possible deterioration in sea-ice conditions

and/or increased iceberg release appears to have taken place over the last ca. 2 cal. ka BP and correspond with lower sea ice occurrence in other sectors of the CAA.

This archive provides a new viewpoint that has enabled us to propose a continuous timeline of the events related to the deglaciation of Kane Basin, which until now relied entirely on land-based studies. Our study suggests that the "bridge dipole" presented in Barber et al. (2001) where warmer (colder) years exhibit more (less) sea ice in Smith Sound and less (more) ice in Nares Strait, may be extrapolated over the last two millennia. Future investigations into the Holocene variability of sea ice conditions in Kane Basin may provide a more comprehensive view on its controlling effect on the NOW polynya. High productivity rates in the NOW Polynya are however also fuelled by the throughflow of nutrient-rich Pacific water via Nares Strait and further investigation into how oceanographic circulation responded to postglacial changes in Nares Strait will provide more insight into the Holocene evolution of this highly productive area of the Arctic. Other than emphasising the need for further research into local reservoir age corrections, our study is inclined to contribute to future work on the export of low salinity Arctic water and Holocene variations of deep water formation (Hoogaker et al., 2014; Moffa-Sanchez and Hall, 2017).

## Acknowledgments

E. Georgiadis' studentship is funded by both the Initiative d'Excellence (IdEx) programme of the University of Bordeaux and the Natural Science and Engineering Research Council of Canada (NSERC). We would like to thank Anne Jennings, Sofia Ribeiro, Audrey Limoges and Karen Luise Knudsen for constructive conversations on the history of North Water Polynya and Benoit Lecavalier for having shared with us the much appreciated Agassiz ice core temperature record. This work is supported by the Fondation Total, the French Agence Nationale de la Recherche (GreenEdge project), the Network of Centres of Excellence ArcticNet and the European Research Council (StG IceProxy). Finally, we wish to thank the CCGS *Amundsen* captain, officers and crew for their support during the 2014 ArcticNet cruise.

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

**Table 1: AMS radiocarbon ages on selected carbonate material. Asterisks indicate data that were not used in the age model. MBF: mixed benthic foraminifera, MS: unidentified mollusc shell.**

| Laboratory code | Dated material | Depth (cm) | 14C age (a BP) | Median probability age (cal a BP) ΔR=0 | 1σ ΔR=0 (cal a BP) | Median probability age (cal a BP) ΔR=240 | 1 σ ΔR=240 (cal a BP) |
|---|---|---|---|---|---|---|---|
| SacA-46000 | MS | 58.5 | 3150+-35 | 2932 | 2869-2984 | 2700 | 2673 - 2736 |
| UGAMS-24304 | MS | 59 | 3125+-25 | 2900 | 2854-2941 | 2683 | 2655 - 2720 |
| UGAMS-24305 | MS | 62 | 3010+-25 | 2775 | 2739-2802 | 2502 | 2428 - 2575 |
| SacA-46003 | MBF | 122 | 6125+-45* | 6555 | 6494-6617 | 6205 | 6259 - 6356 |
| UCIAMS-173009 | MS | 139 | 4540+-20 | 4760 | 4764-4809 | 4427 | 4392 - 4453 |
| UGAMS-24307 | MS | 186 | 5445+-25 | 5823 | 5780-5876 | 5572 | 5541 - 5602 |
| UGAMS-24295 | MS | 207.5 | 6005+-25 | 6417 | 6382-6458 | 6005 | 6168 - 6240 |
| UCIAMS-173006 | MS | 238.5 | 8175+-20* | 8651 | 8595-8690 | 8389 | 8363 - 8411 |
| SacA-46002 | MBF | 251.5 | 7250+-60 | 7714 | 7649-7780 | 7503 | 7451 - 7555 |
| SacA-45999 | MBF | 273.5 | 7870+-50 | 8336 | 8290-8388 | 7870 | 8026 - 8147 |
| Beta-467584 | MBF | 297.5 | 7980+-30 | 8433 | 8388 - 8469 | 8215 | 8167 - 8259 |
| UGAMS-24294 | MS fragment | 301.5 | 43700+-225* | | | | |
| Beta-467583 | MBF | 310.5 | 9380+-30* | 10210 | 10180 - 10234 | 9907 | 9821 - 10001 |
| Beta-467583 | MBF | 327.5 | 8160+-30 | 8633 | 8577 - 8685 | 8379 | 8347 - 8405 |
| SacA-46001 | MBF | 333.5 | 8200+-60 | 8709 | 8587-8796 | 8422 | 8358 - 8482 |
| UCIAMS-173007 | MS | 358.5 | 8450+-20 | 9050 | 9002-9080 | 8703 | 8637 - 8752 |
| UCIAMS-173008 | MS | 362.5 | 8520+-20 | 9149 | 9094-9205 | 8840 | 8773 - 8908 |
| UGAMS-24296 | MS | 407.5 | 8640+-30 | 9318 | 9272-9373 | 8998 | 8968 - 9021 |

| CT-scan | Lithostratigraphic representation | Thin sections | Unit | Description | Sedimentary process | Paleo-environmental implications |
|---|---|---|---|---|---|---|
|  | | | 5B | Silty-clay matrix. Few lonestone. | Hemipelagic sedimentation/ limited contribution of settling from meltwater plumes. Limited ice-rafting. | Glacial distal/hemipelagic sedimentationn. Winnowing from strong subsurface currents. Moderate calving following the deceleration of glacial ice fluxes. Severe sea ice conditions. |
|  | | | 5A | Silty-clay matrix. Less frequent lonestone in comparison to unit 4. | Less deposition from meltwater plumes. Less ice-rafting. | Greater distance of the ice margin to the core site (i. e. retreat of the GIS in eastern Kane Basin). Deceleration of glacial fluxes and/or increased sea ice. |
|  | | | 4 | Dominant clay. Frequent lonestones. | Settling from meltwater plumes. Frequent ice-rafting | Distal glacial marine environment (Ó Cofaigh & Dowdeswell, 2001). Increased calving rates following the collapse of the glacial buttress in Kennedy Channel. Limited sea ice. |
|  | | | 3C | Unsorted silt to gravel/ pebble in a clay matrix. Absence of grading | Iceberg-rafted sediment | Increased calving rates resulting from accelerated glacial fluxes following the collapse of the glacial buttress in Kennedy Channel (MacGregor et al., 2016). |
| | | | 3B | Faintly laminated silty sediment. Lonestones near-absent. | High energy water-transport predominant, minor ice-rafted debris | Entrainment of sediment from northern Nares Strait associated to the establishment of the Hall Basin-Kane Basin connexion through Kennedy Channel. |
| | | | 3A | Unsorted silt to gravel/ pebble in a clay matrix. Absence of grading | Iceberg-rafted sediment | Collapse of the GIS/IIS ice saddle in Kennedy Channel. |
|  | | | 2B | Dominant clay, slightly laminated. Lonestones less frequent in comparison to subunit 2A. | Settling from meltwater plumes. Ice-rafting less frequent. | Distal glacial marine environment (Ó Cofaigh & Dowdeswell, 2001). Increasing sea ice occurrence. |
|  | | | 2A | Gradually finer material. Frequent lonestones. | Settling from meltwater plumes. Occasional ice-rafting. | Growing distance of the ice margin to the core site. Limited sea ice occurrence. |
|  | | | 1C | Unsorted silt to gravel/ pebble in a clay matrix. Absence of grading | Iceberg-rafted sediment | Release of accumulated glacial ice flux following the breakup of sea ice and resulting in intense iceberg calving (Reeh et al., 2001). |
| | | | 1B | Finely laminated sediment. Few or no lonestones | Settling of suspendid sediment from meltwater plumes. Little to no ice-rafting. | Proximal glacial marine environment under severe sea ice conditions (Ó Cofaigh & Dowdeswell, 2001) which may be related to the 9.3-8.2 cold event (Axford et al., 2009; Fisher et al., 2011) |
| | | | 1A | Interbedded coarse and fine laminae. Coarse laminations are occasionally graded | Meltwater plume deposits and small scale pro-glacial debris flows | Ice marginal glacimarine environment (Ó Cofaigh & Dowdeswell, 2001). GIS and/or IIS are close to/at the core site |

**Table 2: Details of CT-scans and thin sections for each lithologic unit of core AMD14-Kane2B and summarised descriptions and interpretations. The paleo-environmental implications discussed in this study have been outlined here.**

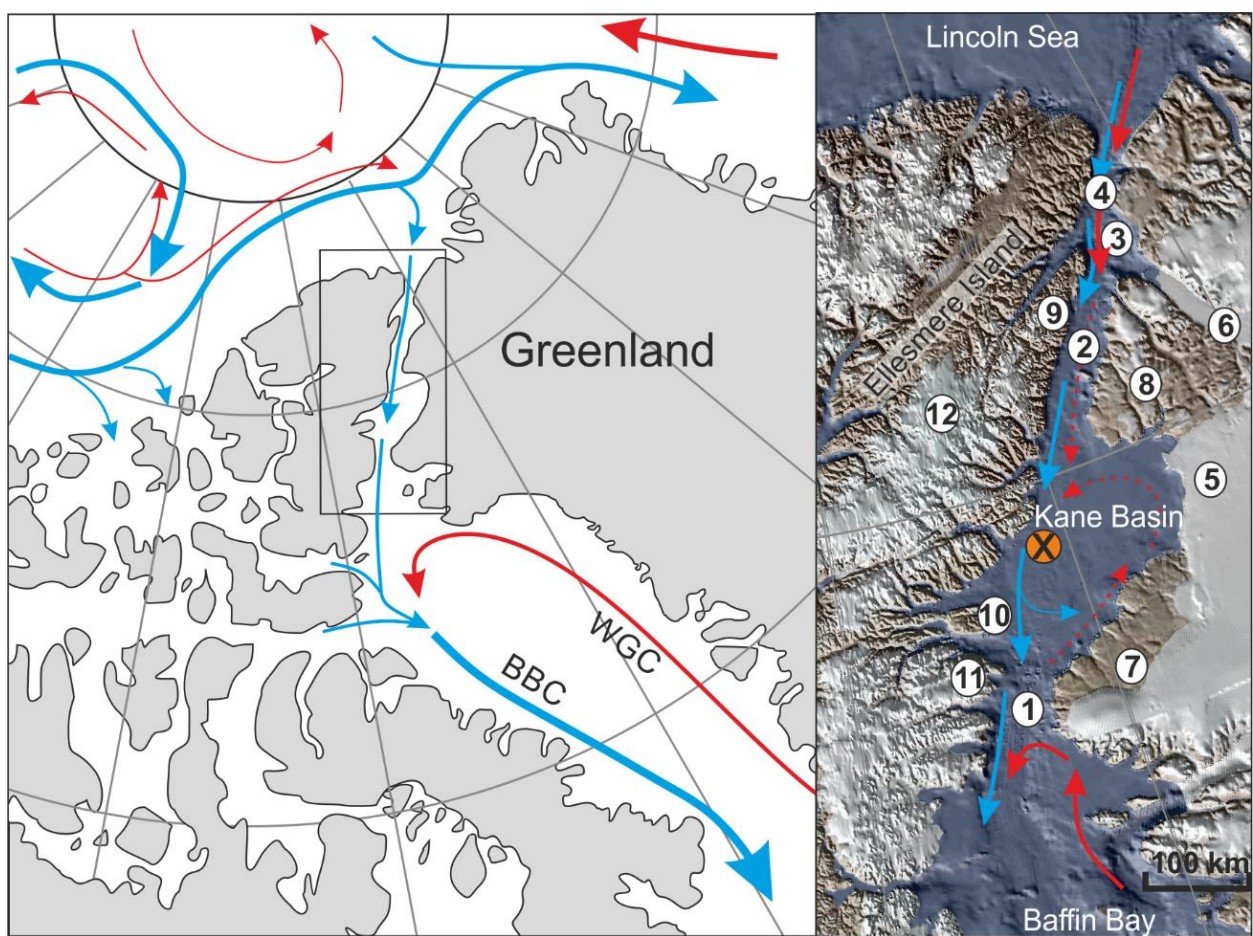

**Figure 1: Schematic circulation in the Canadian and northern Greenland sectors of the Arctic Ocean (left) and within Nares Strait (right). The location of core AMD14-Kane2b is marked by a cross. Blue arrows represent Arctic water and red arrows predominantly Atlantic water. WGC: West Greenland Current, BBC: Baffin Bay Current. 1 - Smith Sound; 2 - Kennedy Channel; 3 - Hall Basin; 4 - Robeson Channel, 5 - Humboldt Glacier; 6 - Petermann Glacier; 7 - Inglefield Land; 8 - Washington Land; 9 - Judge Daly Promontory; 10 - Bache Peninsula; 11 - Johan Peninsula; 12 - Agassiz Ice Cap.**

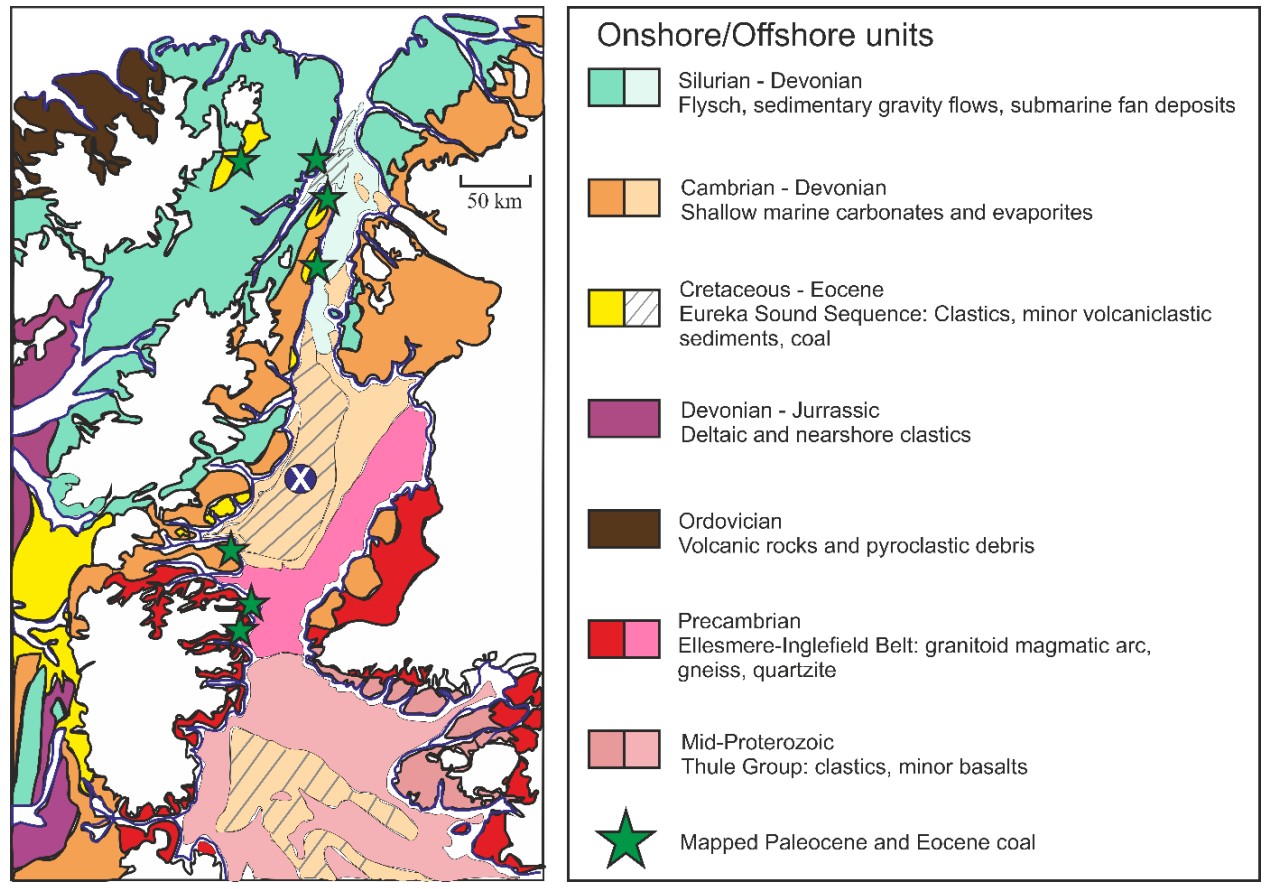

**Figure 2: Geology of Northwest Greenland and Ellesmere Island along Nares Strait. Adapted from Harrison *et al*., 2011.**

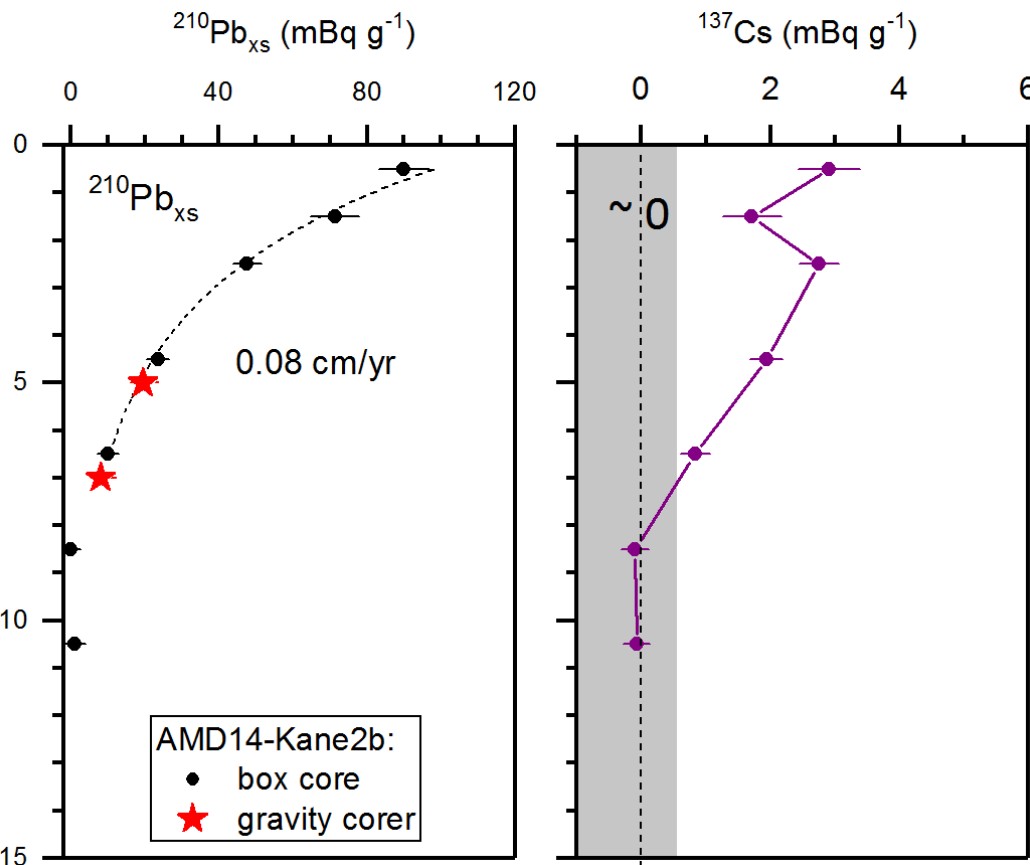

**Figure 3:** $^{210}Pb_{XS}$ and $^{137}Cs$ profiles in AMD14-Kane2b box core (circles). $^{210}Pb_{XS}$ data points in the top part of AMD-Kane2b gravity core have been shifted to obtain the best correspondence of the plots, yielding a material loss of 4 cm at the top of the gravity core.

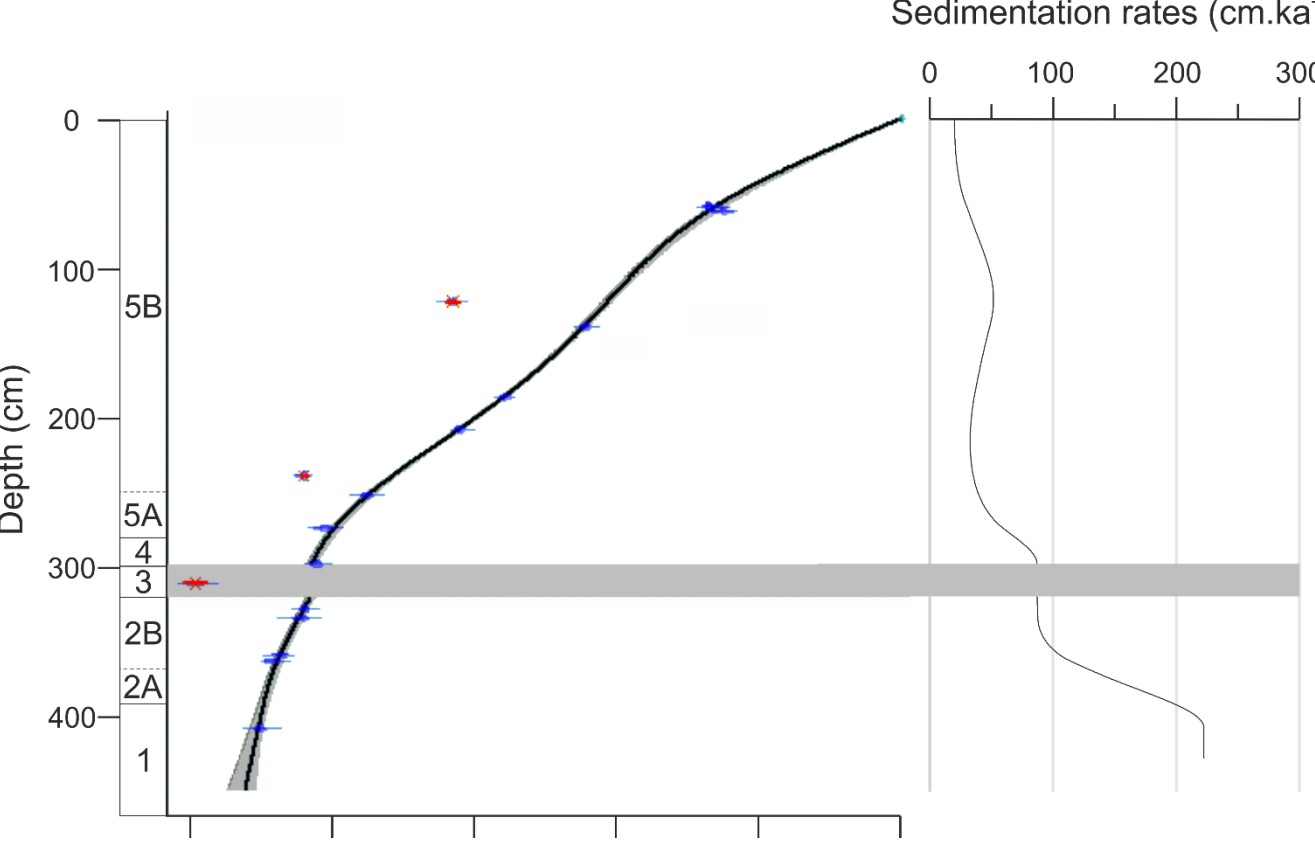

**Figure 4: Core AMD14-Kane2b age model (left) and sedimentation rates (right). The age model is a smooth spline computed using CLAM 2.2 with a smoothing level of 0.4 based on selected radiocarbon dates presented in Table 1. 1σ uncertainty is shown in grey. $^{14}$C ages excluded from the age model (time reversals) are crossed out in red.**

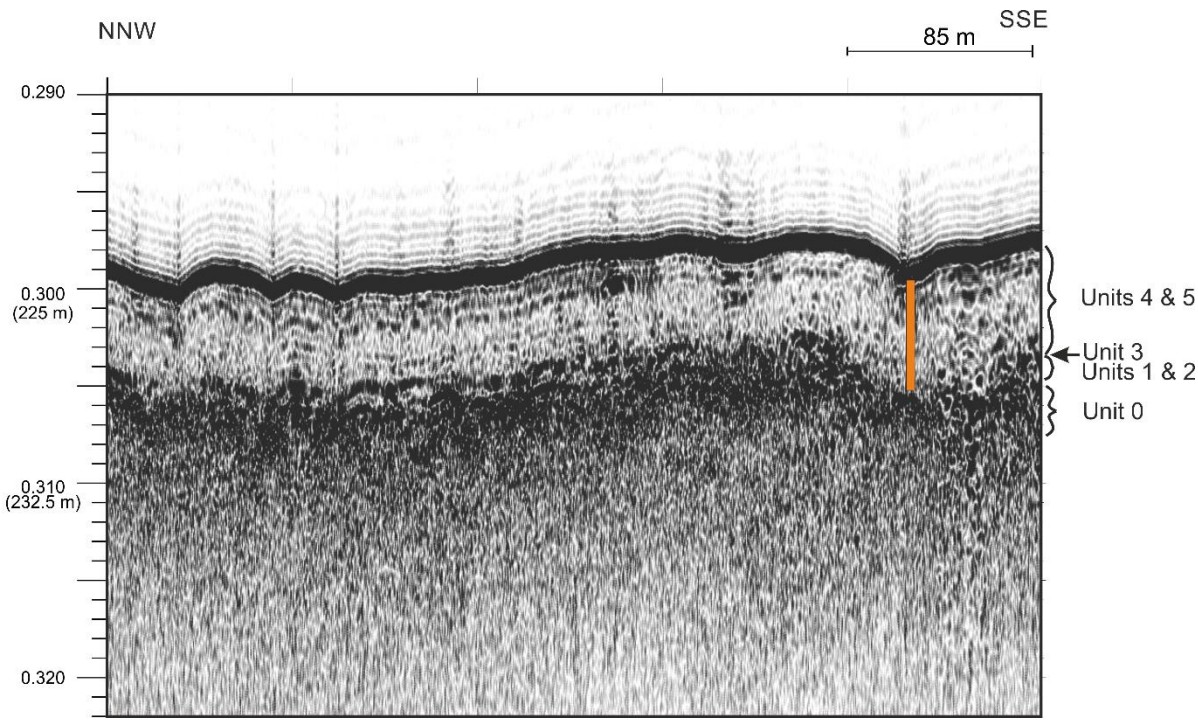

**Figure 5: 3.5 kHz chirp profile across the coring location. Core AMD14-Kane2b is represented by the orange box. Vertical scale in s (TWT) with depth conversion assuming 100 ms (TWT) = 75 meters.**

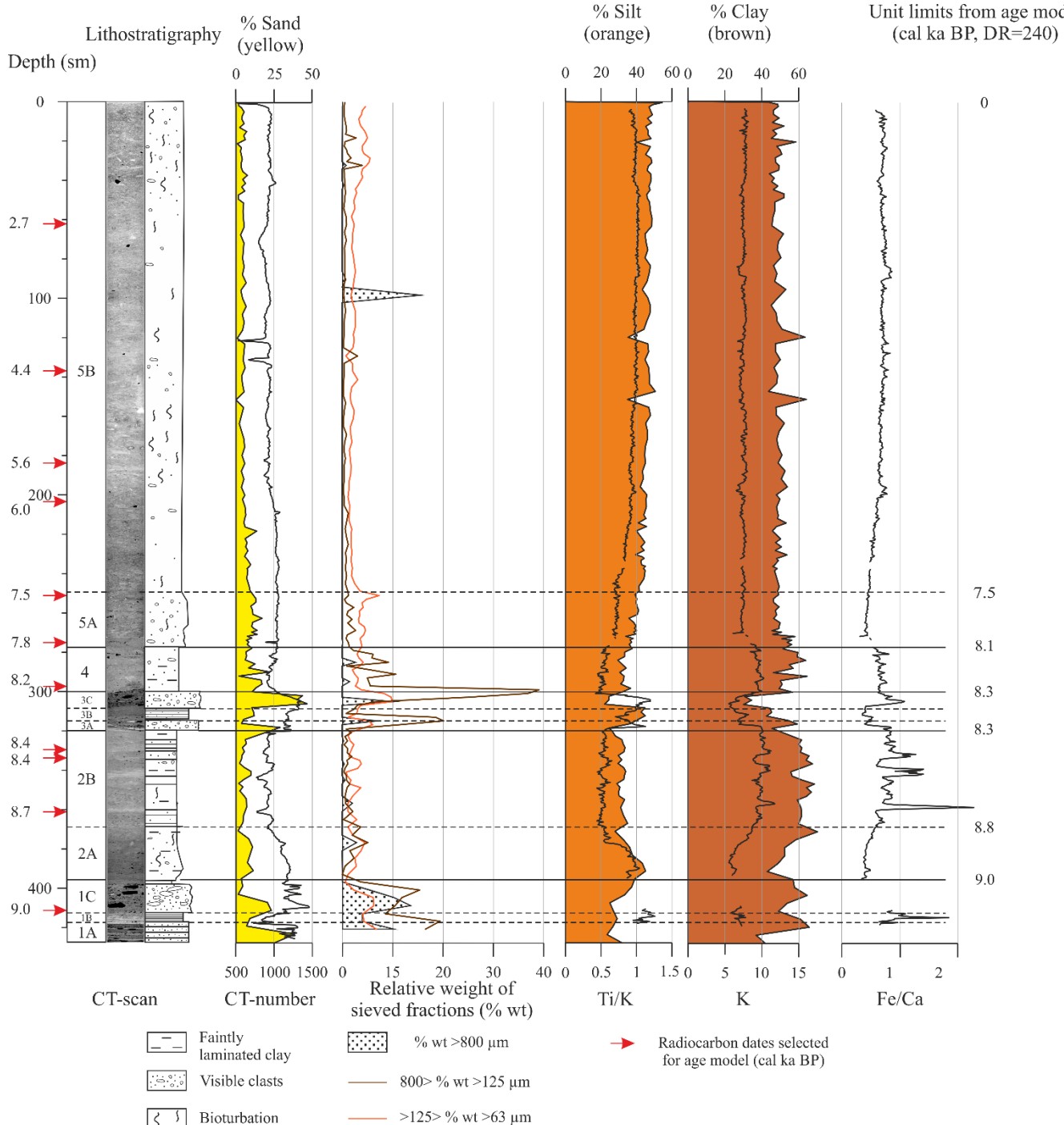

**Figure 6: Sedimentological results and elemental signature of the detrital fraction of core AMD14-Kane2b. Laser diffraction grain size repartition (<2 mm fraction) are shown as % sand, silt and clay. Normalised Zr counts are not shown but their profile is similar to that of Ti.**

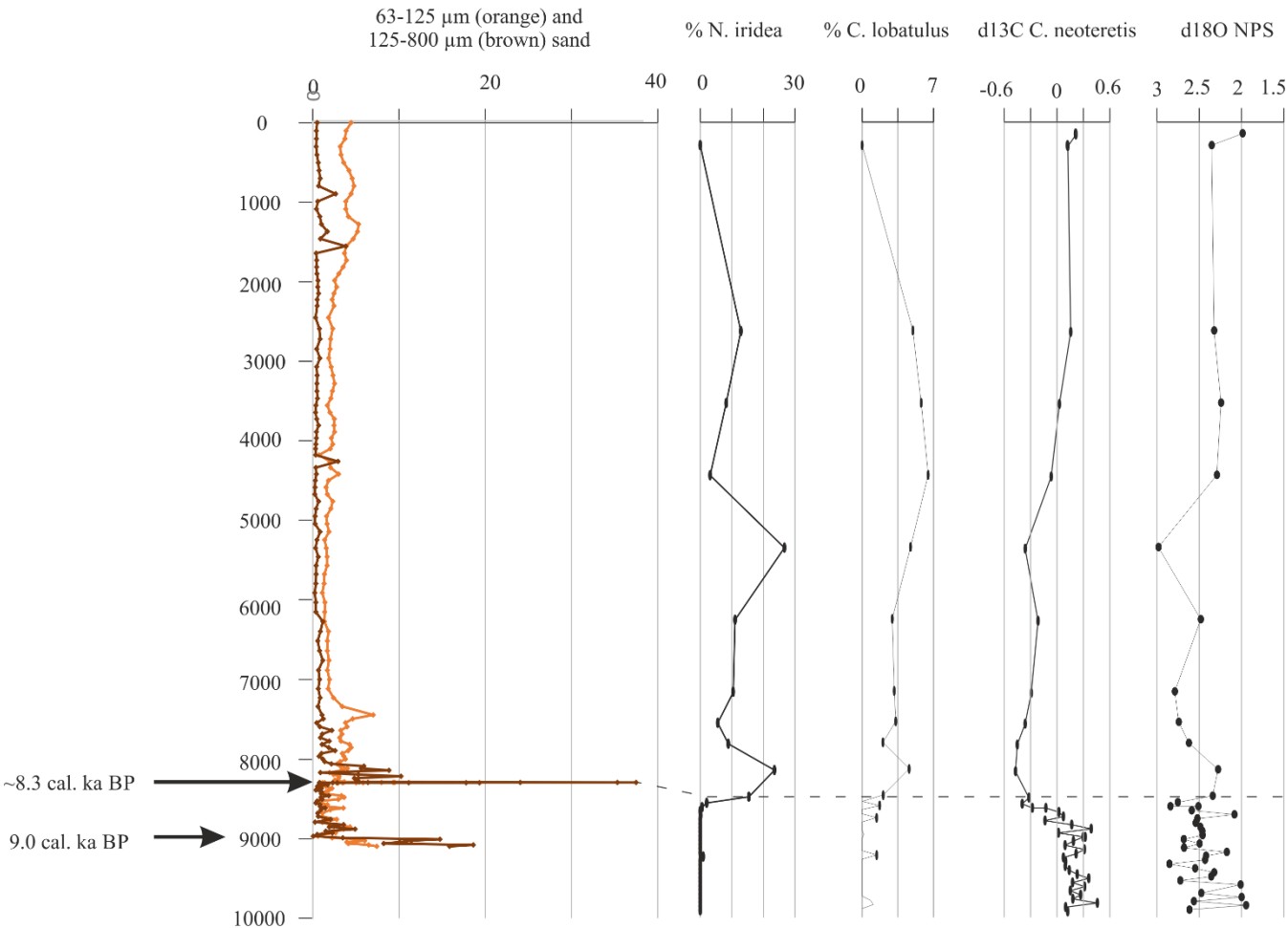

**Figure 7: Comparison of sieved grain size data from AMD14-Kane2b and paleoceanographic proxies from HLY03-05CG in Hall Basin (Jennings et al., 2011). Radiocarbon ages presented in Jennings et al. (2011) were calibrated with ΔR=240 ±51 (Supplementary Figure S.3) years and the age model for core HLY03-05CG is a linear interpolation between the calibrated ages.**

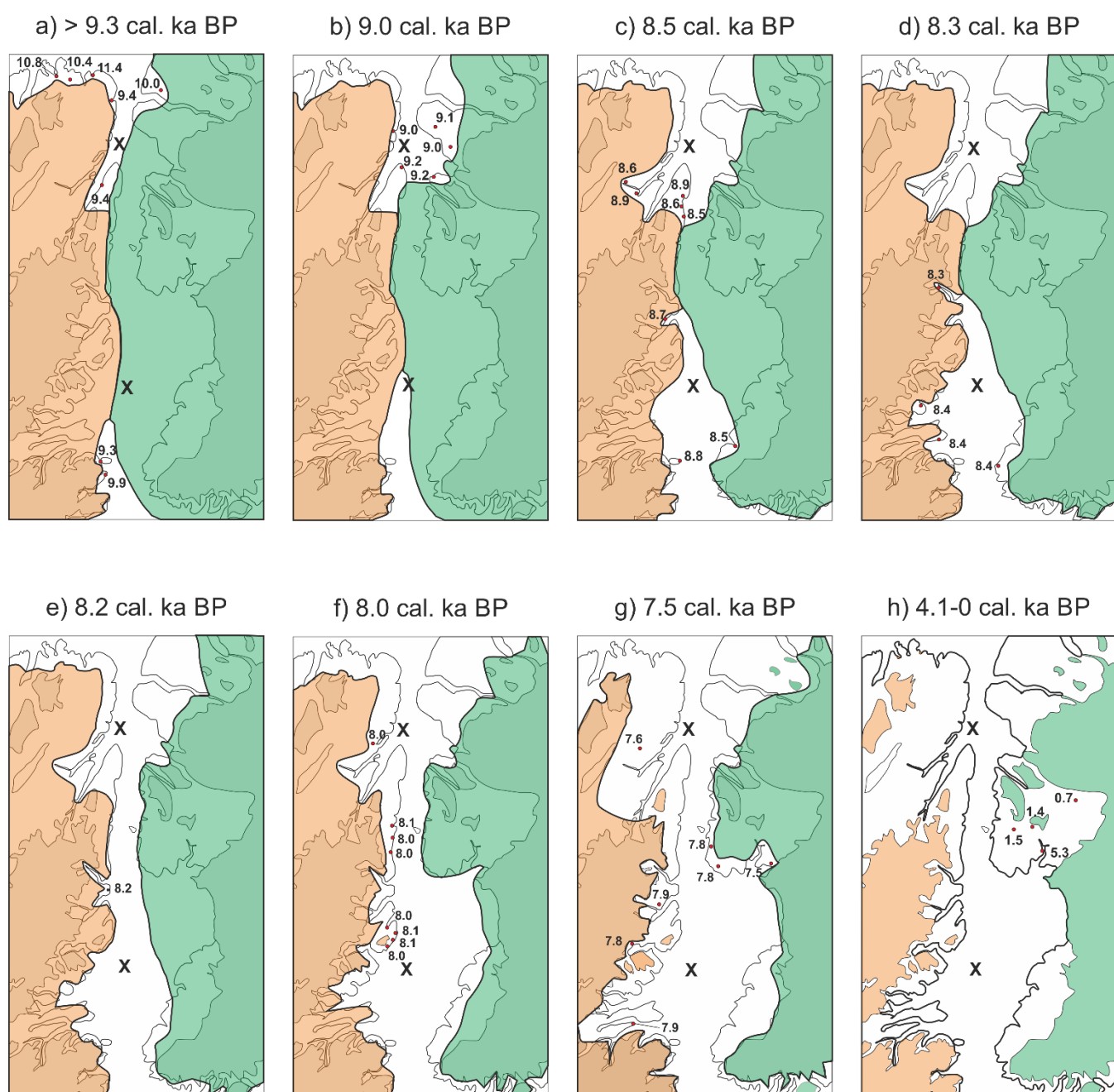

**Figure 8: GIS and IIS retreat in Nares Strait.** Adapted from England (1999) and includes data from Bennike (2002) for Washington Land. Locations for core AMD14-Kane2b in Kane Basin and HLY03-05 (Jennings et al., 2011) in Hall Basin are marked by crosses. All mollusc ages from England (1999) were calibrated with ΔR= 240 using Calib 7.1 (Stuiver et al., 2018) after first adding 410 years to the calibrated ages presented in England (1999) (Supplementary Table S.2 and Supplementary Figure S.3). The position of the GIS and IIS margins offshore in Kane Basin are deduced from our sedimentological and geochemical data, while their locations in Hall Basin are deduced from the data presented in Jennings et al. (2011) and Jakobsson et al. (2018).

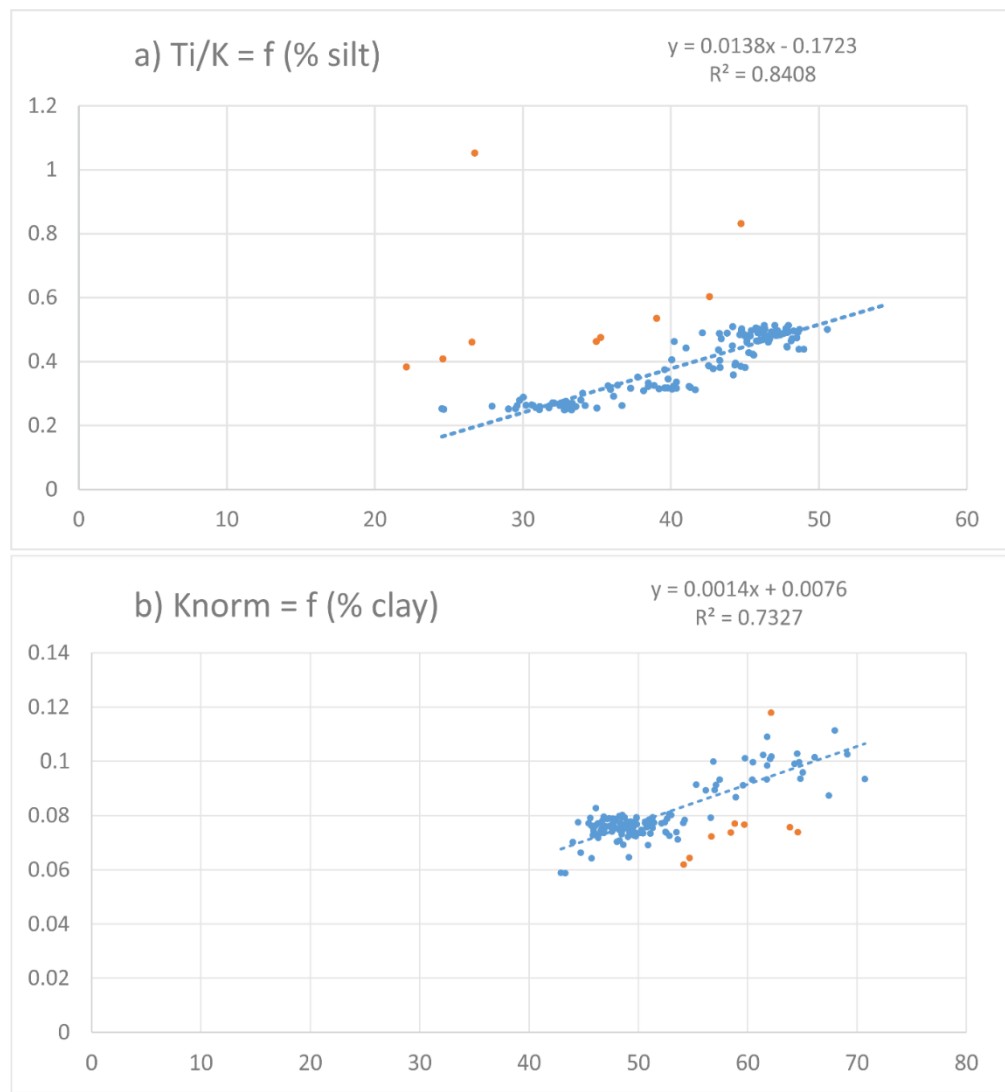

S.1: XRF data plotted against grain size data. a) Ti/K = f (% silt) shows a correlation factor $r^2$=0.84 when 9 outlying data points are omitted (shown in orange). b) $K_{norm}$ = f (% clay) shows a correlation factor $r^2$=0.73 when 9 outlying data points are omitted (shown in orange).

**S.2: radiocarbon ages as reported in England (1999) and Bennike (2002) and calibrated with ΔR=240 ±51 years.**

| | Laboratory dating number | age (yr BP) | 14C age | err | Lat | Long | Median age ΔR=240±51 | sigma1 | Original reference |
|---|---|---|---|---|---|---|---|---|---|
| 1 | GSC-1815 | 10100 | 10510 | 210 | 82°27 | 62°40 | 11386 | 11070 - 11740 | England (1977, 1983) |
| 2 | S-1984 | 9825 | 10235 | 460 | 82°42 | 64°45 | 10781 | 10111 - 11378 | England (1983) |
| 3 | GSC-3744 | 9580 | 9990 | 140 | 82°42 | 68°15 | 10668 | 10483 - 10867 | England (1985) |
| 4 | S-1985 | 9270 | 9680 | 1055 | 82°30 | 64°15 | 10358 | 8991 - 11706 | England (1983) |
| 5 | S-2307 | 9070 | 9480 | 150 | 81°49 | 58°40 | 10010 | 9807 - 10218 | England (1985) |
| 6a | TO-226 | 9010 | 9420 | 150 | 78°36 | 74°45 | 9938 | 9746 - 10159 | Blake (1992) |
| 6b | GSC-2516 | 8940 | 9350 | 100 | 78°36 | 74°45 | 9854 | 9683 - 10028 | Blake (1992) |
| 6c | TO-225 | 8840 | 9250 | 50 | 78°36 | 74°45 | 9681 | 9575 - 9759 | Blake (1992) |
| 7 | TO-136 | 8520 | 8930 | 80 | 81°23 | 66°53 | 9352 | 9274 - 9450 | England (1999) |
| 8 | SI-5551 | 8600 | 9010 | 90 | 82°08 | 62°03 | 9431 | 9345 - 9521 | Retelle (1986) |
| 9 | GSC-3314 | 8470 | 8880 | 100 | 78°43 | 74°43 | 9291 | 9183 - 9427 | Blake (1992) |
| 10 | DIC-737 | 8380 | 8790 | 105 | 81°33 | 64°30 | 9187 | 9036 - 9307 | England (1985) |
| 11a | SI-5855 | 8280 | 8690 | 90 | 81°35 | 60°55 | 9068 | 8963 - 9210 | England (1985) |
| 11b | S-2313 | 8295 | 8705 | 120 | 81°35 | 60°54 | 9082 | 8943 - 9270 | England (1985) |
| 12a | S-1990 | 8255 | 8665 | 215 | 81°53 | 63°20 | 9006 | 8723 - 9289 | England (1985) |
| 12b | GSC-3041 | 8050 | 8460 | 120 | 81°53 | 63°20 | 8746 | 8587 - 8918 | England (1985) |
| 13a | SI-5856 | 8230 | 8640 | 85 | 82°01 | 58°55 | 8994 | 8858 - 9124 | England (1985) |
| 13b | S-2309 | 8205 | 8615 | 135 | 82°01 | 58°55 | 8946 | 8730 - 9132 | England (1985) |
| 14 | SI-5857 | 8225 | 8635 | 95 | 81°18 | 61°21 | 8984 | 8840 - 9128 | England (1985) |
| 15 | DIC-549 | 8200 | 8610 | 260 | 81°15 | 65°45 | 8936 | 8604 - 9252 | England (1983) |
| 16 | GSC-1775 | 8130 | 8540 | 200 | 81°32 | 68°58 | 8850 | 8573 - 9091 | England (1983) |
| 17 | GSC-3286 | 8060 | 8470 | 70 | 78°41 | 74°07 | 8756 | 8626 - 8866 | Blake (1992) |
| 18 | TO-3450 | 8050 | 8460 | 90 | 80°10 | 71°11 | 8744 | 8598 - 8870 | England (1996) |
| 19 | GSC-2843 | 7960 | 8370 | 150 | 81°04 | 66°19 | 8643 | 8425 - 8803 | England et al. (1981) |
| 20 | TO-434 | 7870 | 8280 | 90 | 81°03 | 66°38 | 8505 | 8394 - 8588 | England (1996) |

| | | | | | | | | | | |
|---|---|---|---|---|---|---|---|---|---|---|
| 21 | GSC-3179 | 7860 | 8270 | 270 | 81°41 | 69°08 | | 8549 | 8233 - 8882 | England (1983) |
| 22a | S-2408 | 7825 | 8235 | 130 | 81°46 | 59°08 | | 8472 | 8318 - 8604 | England (1985) |
| 22b | GSC-3693 | 7740 | 8150 | 90 | 81°46 | 59°08 | | 8373 | 8283 - 8474 | England (1985) |
| 22c | S-2301 | 7965 | 8375 | 115 | 81°46 | 59°08 | | 8638 | 8451 - 8775 | England (1985) |
| 23 | L-1091E | 7800 | 8210 | 200 | ~78°38 | ~71°00 | | 8461 | 8194 - 8672 | Nichols (1969) |
| 24 | TO-923 | 7780 | 8190 | 70 | ~78°39 | 71°01 | | 8413 | 8342 - 8484 | Blake et al. (1992) |
| 25 | TO-4192 | 7770 | 8180 | 70 | 79°30 | 74°59 | | 8403 | 8332 - 8474 | England (1996) |
| 26 | S-2109 | 7755 | 8165 | 125 | 81°40 | 65°20 | | 8391 | 8266 - 8535 | England (1983) |
| 27 | GSC-3710 | 7730 | 8140 | 120 | 79°04 | 75°30 | | 8363 | 8233 - 8492 | Blake (1987) |
| 28a | TO-3778 | 7650 | 8060 | 60 | 80°30 | 70°43 | | 8284 | 8218 - 8348 | England (1996) |
| 28b | TO-3464 | 7630 | 8040 | 60 | 80°30 | 70°43 | | 8266 | 8199 - 8328 | England (1996) |
| 29 | TO-3766 | 7540 | 7950 | 70 | 80°13 | 70°08 | | 8176 | 8100 - 8278 | England (1996) |
| 30 | TO-2919 | 7490 | 7900 | 60 | 80°47 | 67°55 | | 8116 | 8032 - 8177 | England (1996) |
| 31 | TO-4210 | 7480 | 7890 | 60 | 79°45 | 71°22 | | 8106 | 8028 - 8168 | Gualtieri and England 1977 |
| 32 | S-2139 | 7385 | 7795 | 375 | 81°41 | 66°21 | | 8042 | 7636 - 8389 | England (1983) |
| 33 | TO-3765 | 7400 | 7810 | 70 | 80°37 | 69°15 | | 8035 | 7955 - 8107 | England (1996) |
| 34a | TO-2922 | 7340 | 7750 | 70 | 80°42 | 68°29 | | 7971 | 7892 - 8046 | England (1996) |
| 34b | TO-2925 | 7620 | 8030 | 600 | 80°42 | 68°29 | | 8337 | 7664 - 8977 | England (1996) |
| 35a | TO-4200 | 7370 | 7780 | 70 | 79°53 | 71°34 | | 8001 | 7925 - 8078 | England (1996) |
| 35b | GSC-5668 | 7320 | 7730 | 80 | 79°54 | 71°30 | | 7950 | 7855 - 8025 | England (1996) |
| 36 | TO-4214 | 7430 | 7840 | 70 | 79°49 | 71°07 | | 8061 | 7987 - 8138 | Gualtieri and England 1977 |
| 37 | TO-4211 | 7390 | 7800 | 70 | 79°41 | 72°17 | | 8022 | 7946 - 8098 | Gualtieri and England 1977 |
| 38 | TO-4198 | 7310 | 7720 | 70 | 80°10 | 71°28 | | 7939 | 7859 - 8005 | England (1996) |
| 39 | GSC-3700 | 7300 | 7710 | 140 | 79°06 | 76°05 | | 7931 | 7782 - 8079 | Blake (1988) |
| 40 | TO-4191 | 7190 | 7600 | 70 | 79°53 | 74°15 | | 7822 | 7755 - 7909 | England (1996) |
| 41 | S-2110 | 6995 | 7405 | 130 | 81°47 | 67°37 | | 7643 | 7517 - 7764 | England (1983) |

| | | | | | | | | | |
|---|---|---|---|---|---|---|---|---|---|
| 42 | SI-3300 | 6860 | 7270 | 70 | 81°17 | 69°25 | 7518 | 7454 - 7573 | England (1983) |
| 43 | GSC-5670 | 6650 | 7060 | 190 | 80°04 | 72°19 | 7322 | 7151 - 7517 | England (1996) |
| 44 | TO-3467 | 6500 | 6910 | 70 | 80°32 | 70°43 | 7199 | 7132 - 7284 | England (1996) |
| 45 | TO-2918 | 6490 | 6900 | 90 | 80°55 | 67°54 | 7184 | 7082 - 7291 | England (1996) |
| 46 | GSC-1614 | 6430 | | 150 | 81°11 | 70°17 | | Driftwood | England (1977, 1983) |
| 47 | GSC-2370 | 6400 | 6810 | 100 | 79°54 | 63°58 | 7079 | 6966 - 7202 | Blake (1987) |
| 48 | GSC-2334 | 5980 | 6390 | 70 | 81°04 | 63°35 | 6582 | 6490 - 6661 | Blake (1987) |
| 49 | GSC-1755 | 6000 | | 150 | 81°04 | 70°00 | | Driftwood | England (1977, 1983) |
| 50a | Beta-91863 | 5920 | 6330 | 60 | 79°09 | 78°13 | 6517 | 6442 - 6594 | England (1999) |
| 50b | GSC-6088 | 5940 | 6350 | 90 | 79°09 | 78°13 | 6350 | 6433 - 6640 | England (1999) |
| 51 | AAR-5768 | 8820 | 75 | 25 | 81°10.6 | 63°20.5 | 9225 | 9409 - 9539 | Bennike 2002 |
| 52 | AAR-5769 | 8010 | 75 | 25 | 81°10.1 | 63°04.9 | 8237 | 8389 - 8539 | Bennike 2002 |
| 53 | AAR-5766 | 6870 | 50 | 24 | 79°55.5 | 64°04.3 | 7162 | 7328 - 7427 | Bennike 2002 |
| 54 | AAR-5762 | 7240 | 65 | 23 | 79°56.5 | 64°17.1 | 7495 | 7636 - 7775 | Bennike 2002 |
| 55 | AAR-5755 | 6410 | 55 | 22 | 80°05.8 | 64°39.4 | 6605 | 6810 - 6961 | Bennike 2002 |
| 56 | AAR-5758 | 7090 | 80 | 21 | 80°24.0 | 66°58.2 | 7364 | 7496 - 7640 | Bennike 2002 |
| 57 | AAR-5757 | 7570 | 65 | 20 | 80°12.6 | 67°11.9 | 7793 | 7957 - 8102 | Bennike 2002 |
| 58 | AAR-5761 | 6890 | 60 | 19 | 80°21.5 | 67°18.7 | 7181 | 7338 - 7458 | Bennike 2002 |
| 59 | AAR-5760 | 7580 | 55 | 18 | 80°18.7 | 67°23.6 | 7803 | 7972 - 8103 | Bennike 2002 |
| 60 | AAR-5755 | 5165 | 55 | 19 | 80°08.8' | 64°20.2' | 5255 | 5470 - 5578 | Bennike 2002 |
| 64 | AAR-5772 | 1400 | 60 | 6 | 80°33.1' | 67°11.1' | 712 | 892 - 1027 | Bennike 2002 |
| 61 | K-7142 | 1310 | 35 | 15 | 80°09.4' | 63°39.6' | 638 | 609 - 672 | Bennike 2002 |
| 62 | K-7138 | 2170 | 55 | 38 | 80°23.9' | 65°18.1' | 1477 | 1693 - 1834 | Bennike 2002 |
| 63 | AAR-5531 | 2070 | 55 | 39 | 80°24.9' | 64°20.0' | 1376 | 1563 - 1706 | Bennike 2002 |

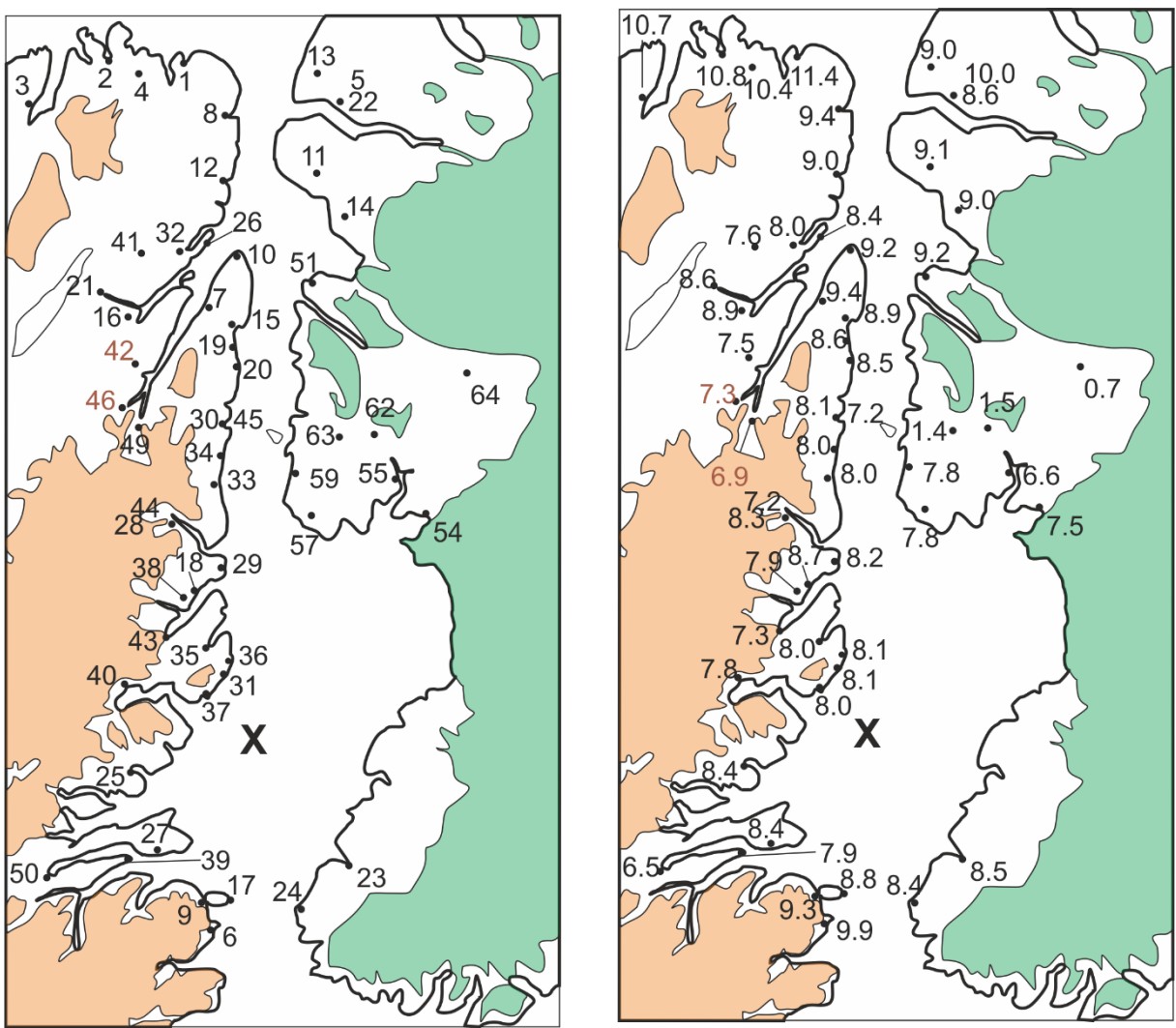

**S.3: location of the radiocarbon ages in Nares Strait reported in England (1999) and Bennike (2002) and their calibrated with ΔR=240 ±51 years.**

**S.4: radiocarbon ages from Jennings et al. (2011) calibrated with ΔR=240**

| Depth in core (cm) | Laboratory number | 14C age | Material dated | Median age (ΔR=240) | 1σ ΔR=240 |
|---|---|---|---|---|---|
| 0–2 | AA-81309 | 530 ±53 | *Bathyarca glacialis* | ~290 | |
| 8–10 | NOS -71686 | 3100 ±35 | NPS | 2636 | 2595 - 2709 |
| 28-30 | NOS -71687 | 5040 ±40 | NPS | 5087 | 5010 - 5140 |
| 58-60 | NOS -71688 | 6870 ±45 | NPS | 7164 | 7120 - 7234 |
| 68-70 | AA-81310 | 7302 ±61 | NPS | 7543 | 7484 - 7596 |
| 69-98 | NOS -72574 | 8290 ±50 | NPS | 8502 | 8439 - 8558 |
| 345-349 | NOS -71689 | 9320 ±45 | NPS and C. neoteretis | 9794 | 9702 - 9882 |