# Peer review of "Deglacial to postglacial history of Nares Strait, Northwest Greenland: a marine perspective from Kane Basin"

_Climate of the Past, 2018_

## Referee Comment (RC1) · Anonymous Referee #1 · 8 Aug 2018

Thank you for the opportunity to review Georgiadis and colleagues paper. They present new data, including grain-size, CT, XRF, and radiocarbon, from sediment core AMD14-Kane2b from Kane Basin and discuss implications for the deglaciation of Nares Strait. They infer a major deglacial event, the opening of Nares Strait, from an IRD event and XRF geochemistry. It is a good dataset and is suitable for publication in Climates of the Past. However, there are a few important issues in the discussion and data treatment that should be addressed before this manuscript is accepted for publication.

First, I like to praise how the paper focuses on a detailed description of the core stratigraphy on depth. The inclusion of Table 2 in addition to Figure 5 make it very easy for me, the reader, to understand the stratigraphy of the core and the author's interpretation of that stratigraphy. In my view, the most important take away from this paper is

the clear description of the stratigraphy and I applaud the authors for that.

My biggest issue with the paper is how the authors make statements regarding the meaning of the data and then fit their interpretations to that model. This is particularly true for the XRF data. With the detailed grain-size data set the authors have generated, it would be much more informative to learn about the relationship between sediment geochemistry and grain-size based on Kane Basin data instead of importing conceptual models from vastly different depositional environments. This would make the results of this study much more convincing and help other researchers working in the region. At a minimum, the authors need to be clearer about what is their interpretation and what is supported by data in the results section. I recommend adding a figure showing the relationships between XRF element counts and particle size in the various lithologic units, as this relationship (or lack of relationship) is central to many of the interpretations made by the authors.

I would also like to see the authors expand their discussion to include how their data compare to another marine perspective on the Holocene deglaciation of Nares Strait by Jennings et al. (2011). Although the paper is referenced in the introduction and the discussion, the authors do not address why their age for the opening of Nares Strait is younger. I believe the two observations can be reconciled, but it is worth a discussion by the authors as Jennings et al. present faunal and stable isotope data that clearly show the change in oceanographic conditions with the opening of the strait and have a high quality age constraint above the transition at 8,328-8,528 cal yrs BP ($\Delta$R = 335 $\pm$ 85) based on Neogloboquadrina pachyderma sinistral. I consider these to be more reliable evidence than semi-quantitative bulk-sediment geochemistry and an IRD event layer.

I have included specific comments related to each section and line comments for each section below.

Section/Line comments:

Title: As a suggestion, perhaps include the geographic location of your study site, such as "a marine perspective from Kane Basin," to draw more attention to your article from researchers working in or interested in Kane Basin.

Abstract: The abstract is fine and includes the major interpretations made by the authors. My only general recommendation, beyond the line comments would be to include qualifiers like around before the dates. For example, changing "at 8.3" to "around 8.3" and "from 7.5 to 1.9" to "from about 7.5 to 1.9." There is a great deal of uncertainty in your age model and it is always good to indicate that these are just estimates of the events, especially when you are not reporting the uncertainty associated with those event ages (which would be difficult to quantify with a program like CLAM).

Page 1 Line 12: For clarity, perhaps rewrite "that translate into ice sheet configuration in the strait" to "that provide new insight to the ice sheet configurations in Nares Strait."

Page 1 Line 14: It is not clear, even after reading the paper, what you mean by "unstable sea surface conditions." It would be helpful to clarify what this means and maybe what data it is based on.

Page 1 Line 20: Change sediment source to sediment geochemistry

Introduction: This is a nice introduction and clearly introduces the problem you are trying to address.

Page 2 Line 4: I recommend starting a new paragraph after discussing the modern observations and beginning the discussion of the Holocene observations.

Page 2 Line 6: I would suggest changing the word admitted. "It is now widely admitted" reads awkward

Page 2 Line 8: You could also include the palaeoceanographic evidence from Hall Basin (Jennings et al., 2011).

Page 2 Line 19: Include the name of the core

Page 2 Line 19: Perhaps also indicate that you are presenting radiocarbon data. For example, "sedimentological, geochemical, and geochronological data. . ."

Regional Settings: This is a nice overview of the regional oceanography and geology. I would just be careful about making the jump from regional geology to geochemical signatures for source regions. While it makes sense that Ca is probably from the carbonate rich regions and the K is from gneiss regions, we don't have a good sense from your paper (or previous work) what the sub-ice bedrock is or what the geochemical variability of these units (like the siliciclastic sedimentary rocks found throughout the region). I would also include a call out to your Figure 2 when you discuss the regional geology.

Materials and Methods: The CT, grain-size, and XRF methods all seem appropriate and sufficiently described. However, as noted in the line comments, I do not find the conceptual framework for the XRF interpretation convincing and needs further support from data and/or references. The grain-size statement would fit much better in the results section, after a more direct comparison of the 'grain-size sensitive elements' to the detailed grain-size record the authors have generated. This would be the first direct observation I am aware of for Kane Basin or Nares Strait and would be a helpful observation for others working in the region.

The choice of $\Delta R$, 240 years, seems reasonable and is similar to the recent Jakobsson et al. (2018) choice of $268 \pm 82$ years. I would have liked to see uncertainty included in the calibration of the dates, as the authors acknowledge the pre-bomb $\Delta R$ range could have been as high as 335 years, but I doubt that would have changed the overall interpretation, as the uncertainty with respect to the material dated in this study is likely larger than the uncertainty associated with the choice of $\Delta R$ (with 35% of the reported ages rejected in the age model). Given this uncertainty, perhaps it would be better to present the chronology in the results section, after discussing the radiocarbon results and specific justification for which ages you accept/reject.

Page 4 Line 5: Change section number from 2 to 3. Likewise, subheading numbers need to be changes from 3.2 to 3.1 and 3.3 to 3.2

Page 4 Line 8: Does u-channel need to be capitalized?

Page 4 Line 8: Define CASQ acronym

Page 4 Line 11: Define INRS acronym

Page 4 Line 27 to Page 5 Line 2: The authors need to do a better job justifying their interpretation of the XRF data. The sentence that begins "The sedimentary rocks from northern Kane Basin and Nares Strait. . ." needs to be supported by data or a reference. I am not convinced that these are the geochemical signatures of each source region otherwise. I also do not think it is convincing based on the low resolution geologic map presented as Figure 2. Additionally, the second sentence that begins "Additionally, heavy elements Ti and Zr reflect the grain size variation. . ." is often true; however, Ti and Zr can also be influenced by provenance changes. The cited references for this statement involve study areas on the Oman Margin, NW Mexican Margins, and Gulf of Cadiz (and a fourth by Correns, from 1954, which is not listed in the reference list)—not Kane Basin. I could list other studies from different regions which show Zr variations are not related to lithic particle size, such as Phillips et al. (2014) in the Bay of Bengal. The point I am making is that I would like to see Georgiadis and colleagues make observations based on the data set they have generated from Kane Basin, rather than fitting their data to a preconceived notion. I would also argue that the authors are in a great position to do exactly that, as they have generated a detailed grain-size record to go along with their XRF geochemistry.

Page 5 Line 7: The supplementary figure could be moved to the main text, as the manuscript only has six figures. They are good data–why bury them in a supplement when you don't need to?

Page 5 Line 29: Indicate what settings you used in CLAM 2.2 (i.e. is this a polynomial

fit? Or linear interpolation?)

Page 5 Line 31 to Page 6 Line 3: This paragraph would be a better fit for the results section after a discussion of the radiocarbon results and specific criteria for rejected dates.

Results and Interpretations: As stated earlier, I think the authors do a great job discussing the stratigraphy of the core, the depositional processes that may be important in creation of the deposit, and, for the most part, saving the narrative for the discussion.

Page 6 Line 12: Can you say anything about the thickness of unit 0 given that the acoustic waves are likely attenuated? I think your interpretation is correct, I just don't think you have any evidence for the thickness of the unit.

Page 6 Line 17: Is the evidence for 'The elemental composition of the sediments is largely dominated by Ca throughout the core' based on the XRF data? If so, even after your normalization, semiquantitative count differences between elements don't necessarily mean weight percent differences. I think you would need a more quantitative method and/or a calibration to make that statement.

Page 6 Line 18: Why does K reflect source changes and not particle size changes. This needs to be explained further and supported by data and/or references.

Page 7 Line 4-5: Perhaps clarify that the larger particles are relative to overlying units, as it seems Unit 1B is finer grained than Units 1A and 1C.

Page 7 Line 27-30: Indicate that these are likely sources or your interpreted sources.

Page 8 Line 5: I would change "A significant portion of these sediments derived from eastern Kane Basin gneisses" to something like "Relatively high K counts likely reflect an increase contribution from eastern Kane Basin gneisses."

Page 9 Line 22-25: Like other comments, at this point state the geochemical evidence for your interpretation. The contribution of northern and/or eastern carbonates doesn't

increase slightly, Ca counts increase slightly which is likely consistent with an increased contribution of northern and/or eastern Kane Basin carbonates.

Page 9 Line 27: What is Unit E? Do you mean Unit 5?

Page 10 Line 5-6: Make sure that you are discussing the data. There wasn't a change in the delivery of gneiss material. There was a change in K and Ca, which you interpret as a change in provenance. But as you have already discussed, there is also a big change is sediment grain-size.

Page 10 Line 15: The authors indicate that two mollusk shells are younger than expected and claim that they were likely remobilized by bioturbation. In one case this would indicate movement of about 80 cm downward in the core (UGAMS-24308). Is this a reasonable interpretation for a mollusk shell? Are there any references that could be used to support that interpretation? Would the simpler interpretation be to accept the younger dates and attribute the older dates to reworking, as you do for the older part of the age model?

Discussion The review of the terrestrial data and comparison to your core data is good. As stated in the intro, I think this paper is missing a comparison to the other important marine perspective–from Hall Basin. It is also my opinion that many of the interpretations are a bit speculative in the discussion section, such as the influence of the 9.3 ka cold event, but I think, because it is in the discussion only, it is okay. However, it detracts from one of the main and well-supported findings of the paper (in the same section), that your region of Kane Basin was deglaciated by 9 ka and that provides an important constraint for your Figure 6. In this instance and others, you might consider focusing your discussion on the major and best supported findings so that they are the main focus of the reader's attention.

Page 11 Line 16: Not necessary for this paper as it was only available online after you submitted your paper to Climates of the Past, but you may also be interested in a recent cosmogenic study by Reusche et al. (2018) for your future work.

Page 11 Line 24: Do you mean Figure 5?

Page 13 Section 5.3: I do not find the presence of old foraminifera convincing evidence for a Northern Nares Strait sediment source. Obviously, there is something complicated happening with radiocarbon, as 35% of your reported ages are rejected and not used in your age model. Part of your argument may be that these are on mixed benthic forams; however, you have other mixed benthic forams which are clearly too old elsewhere in you core (e.g. SacA-46003) without similar sedimentological indicators. I think this part of the discussion could use a little work and dive a bit more into the uncertainties of your chronology and the constraints from the Hall Basin record (and its uncertainties). Page 13 Line 32 and Page 14 Line 4: You use the term 'gneiss signal.' As I've been critical of in other parts of this review, you present no data that K is a proxy for gneiss or sediment sourced to a specific region.

Conclusions:

Page 15 Line 17: Change section number from 5 to 6.

Page 16 Line 2: Authors claim this is the first non-land view point of the deglaciation of Nares Strait. This is not true, as a northern perspective of the events was documented by Jennings et al. (2011) using a marine sediment core from Hall Basin. The authors need to change this to a southern perspective, or a Kane Basin perspective.

Figures:

Figure 3: In the caption, indicate that the age model is a polynomial, spline, or linear interpolation (whatever you used) and indicate if the shading is the one sigma or two sigma uncertainty of that model fit.

Figure 5: In the caption, indicate that the plotted radiocarbon dates are only the ones you accepted to use in your age-depth model and make a call out to Table 1, so it is clear to the reader that there are other radiocarbon data.

References Cited in Review:

Jakobsson, M., Hogan, K.A., Mayer, L.A., Mix, A., Jennings, A., Stoner, J., Eriksson, B., Jerram, K., Mohammad, R., Pearce, C., Reilly, B., Stranne, C., 2018. The Holocene retreat dynamics and stability of Petermann Glacier in northwest Greenland. Nat. Commun. 9, 2104. https://doi.org/10.1038/s41467-018-04573-2

Jennings, A.E., Sheldon, C., Cronin, T.M., Francus, P., Stoner, J., Andrews, J., 2011. The Holocene history of Nares Strait: Transition from glacial bay to Arctic-Atlantic throughflow. Oceanography 24, 18–33.

Phillips, S.C., Johnson, J.E., Giosan, L., Rose, K., 2014. Monsoon-influenced variation in productivity and lithogenic sediment flux since 110 ka in the offshore Mahanadi Basin, northern Bay of Bengal. Mar. Pet. Geol., Geologic implications of gas hydrates in the offshore of India: Results of the National Gas Hydrate Program Expedition 01 58, 502–525. https://doi.org/10.1016/j.marpetgeo.2014.05.007

Reusche, M.M., Marcott, S.A., Ceperley, E.G., Barth, A.M., Brook, E.J., Mix, A.C., Caffee, M.W., 2018. Early to Late Holocene Surface Exposure Ages From Two Marine-Terminating Outlet Glaciers in Northwest Greenland. Geophys. Res. Lett. 0. https://doi.org/10.1029/2018GL078266

---

## Author Comment (AC1) · 9 Aug 2018

Dear referee,

We would like to thank you for having provided us with such an early review of our work along with an opportunity to revise it. The points that you address are very pertinent and we will take them into account in the revised version of our paper.

We will make sure to use our grain size data to better interpret the XRF data: there is an excellent correlation between normalised Ti counts and silt contribution in the <2 mm fraction. Better assessing the local variability in the geological sources that are prone to inducing changes in our elemental dataset may prove challenging, but we believe that our interpretations of the XRF data in terms of sources, though preliminary, reflect

general trends in the sediment provenance. Caron et al. are working on a forthcoming paper with XRD data on the same core which will bring further insight into the Holocene sediment sources in Kane Basin and will be able to assess the reliability of the time-efficient XRF approach that we use. We will specify this in the revised version of our paper. We will also revise the age model of the top part of our core using the more logical younger ages, and incorporate the important references you mention in our discussion, such as the opening of the Strait as viewed from Hall Basin (Jennings et al., 2011).

We are glad to welcome the opportunity to submit a revised version of our paper and will do so as soon as we receive all the reviews.

Yours sincerely,

Eleanor Georgiadis on behalf of the authors

---

## Referee Comment (RC2) · Anonymous Referee #2 · 10 Aug 2018

I have read this manuscript with great interest. It uses data collected from a single, strategically located core from Kane Basin to infer changing paleoenvironmental conditions in response to ice sheet recession in Nares Strait from early Holocene to present. The paper is clearly written and well organized. The main data include lithofacies descriptions from visual core descriptions and CT scanning, CT number as a measure of density, grain size data to infer sedimentation processes, and XRF data to infer sediment provenance. The core chronology is based on 13 radiocarbon dates on benthic foraminifera and molluscs. A local reservoir correction of 240 years was applied to calibrate the dates.

Figure 4, the 3.5 kHz chirp profile over the core site shows that the core penetrates the section essentially to acoustic basement which is interpreted to be subglacial till at the

[Figure]

site, meaning that the core should contain a sequence that includes initial deglaciation, opening of Nares Strait, and continued ice recession from the ocean all along the strait.

I think that the interpretation of the events in Kane2B is reasonable, but there is an alternative interpretation of the sequence of events that also should be considered. In part this stems from a new article that includes new information about early Holocene moraine ages of the Humboldt Glacier:

Early to Late Holocene surface exposure ages from two marine‐terminating outlet glaciers in northwest Greenland Melissa M. Reusche, Shaun A. Marcott, Elizabeth G. Ceperley, Aaron M. Barth, Edward J. Brook, Alan C. Mix and Marc W. Caffee Accepted manuscript online: 13 JUL 2018 12:00AM EST | DOI: 10.1029/2018GL078266

In this paper surface exposure ages indicate that an early Holocene lateral moraine of the Humboldt Glacier was abandoned by 8.3±1.7 ka. Could retreat from this stable position be the source of the increased IRD and other features of unit 3A,B,C rather than being a signal of the opening of the strait? Alternatively, the retreat from the moraine might signal the ice recession that lead to opening of the strait, which would support the argument made in this paper. A connection between retreat from the moraine and opening of the strait was not made by Reusche and others, mainly because in the literature the opening of Nares Strait is suggested to be earlier by some hundreds of years.

An alternate scenario for the stratigraphy of the core is: Unit 1, ice proximal, reflects opening of Nares Strait and Unit 3 represents retreat of the Humboldt Glacier.

Given the uncertainties of the reservoir corrections used all along Nares Strait and through the Holocene, I think it would be useful for the authors to consider this alternative scenario. If the benthic DR in Kane Basin is larger than the 240 years used, then the deeper unit 5 could represent opening of the Strait, which would mean that all of Kane 2B was deposited after the strait opened. Can other arguments about environmental changes one might expect from the opening of the strait be brought to

bear to support one interpretation or the other? It is clear that the age presented for the opening of Nares Strait by Jennings et al. (2011) is somewhat older than 8.3 ka, even when the higher DR of 335 years is used. In any case, I think it is reasonable to consider an alternative interpretation of the events. In the end, one can favor one interpretation over the other based on the evidence. But I don't think the origin of Unit 3 can be proven with the data shown, especially given the uncertainties in the reservoir corrections and the short time span involved. However, it would be really interesting if the moraine recession, Unit 3, and the opening of Nares Strait were linked.

Minor comments:

P1. Line 8. 8 ka 14C calibrates to 8.7 cal with DR=240 I think. I think the other calibrations of dates from the literature are done correctly with the reservoir correction added back before the dates were calibrated. But it is worth checking. Especially with regard to Figure 6. I missed a description of how the dates were recalibrated. And there might need to be a list of the dates shown on the maps with the 14C age and the cal age along with the references etc.

P2. Line 18. What about St. Onge and St. Onge, 2014?

P3, 6. punctually...not sure what you mean.

P3, 7,8. Make this 2 sentences.

P3, 15. Ice arches only block the sea ice...and allow the liquid freshwater through.

P3, 17. change 'associated to' to 'outlets of'?

P4, 16, 28. Calculate and use the $\pm$ on the DR.

P4, 31. Change 'drastic' to 'major'?

P6, 7. You may be overinterpreting these thin units. Change 4 to 6 cm. They all appear to be ice proximal and there is variability in the sedimentation along an ice margin that can be reflected in the sediments you have.

P6, 18. Change 'oscillates' to 'varies'?

P6, 23. Evolve, decrease P6, 24. Meltwater plumes and iceberg rafting. P8, 20. And P13, 6. What species were dated? Can these be a mixture of >50 k and contemporaneous forams? It seems unlikely that forams would look fresh after being entrained in a current. This is not very convincing about forams coming from Hall Basin. P9, 1. along with P9, 11. Suggest you delete 'discretely'. P9, 21. Svendsen P9, 25. Keep the format you were using with the unit name and details at the start of the line. P9, 27. Unit 5 rather than E. P10, 3. Suggest you skip the double negative and say. . .'We rule out hypothesis 3. . . P10 in general. It would be helpful to name the species dated. . .both mollusk and foram when possible. There are some species to avoid and it is useful to have the info.

P11, 6,7. I don't think the second part of this sentence is right. Suggest you end sentence at 'strait'. And delete the rest except keep the reference.

P12, top paragraph. I think this part is over interpreted. And there is quite a bit of >800 micron material. Basically this unit just seems ice proximal and does not necessarily suggest the sequence of environments.

P12. Discussion of Fig. 6. How about putting on a through h designations on your panels. Then you can refer to the right panel in your discussion. P12, 20. Change 8.3 to 8.5. 8.5 is what is on the map.

P12, 32. Coarse fraction content is not only influenced by sea ice. I think this is overinterpreted.

P13, 32. Sea ice and ice bergs certainly are not mutually exclusive. I think you are over interpreting sea ice with your grain size data because you have a lot of coarse material in your units. Icebergs carry all grain sizes.

P14, 5. Outset change to onset?

P14, 8. A stretch for you to interpret sea ice here with just the grain size.

P16, first para. Your interpretation is not unique. . .there is at least one other alternative. It is best to explore that. The DR is poorly constrained.

Fig. 3. Add tick marks to the axes. Add the lithofacies units to the depth scale. Fig. 4. Can you remove the interpretation and put it in a panel below? One cannot see the boundaries on the data. Figure 5. why is lithofacies 1 2 boundary listed as 9.1 when an underlying age is 9? Fig. 6. Explain how you calibrated the ages in the map and list the ages in a table with their pertinent information.

Table 2. fix the tick marks and add the date levels where they coincide with the examples of lithofacies.

---

## Author Comment (AC2) · 11 Aug 2018

Dear referee,

We are very grateful that you provided us with a review of our work ahead of the deadline and that you allow us the opportunity to submit a revised version.

Thank you for pointing out Reusche et al.'s recent paper that we were not aware of at the time of writing our manuscript. Given the scarcity of archive in Nares Strait, we welcome this new study which we must of course take into account in our work. The remarkable coincidence in the dating of the retreat of the Humboldt Glacier from a lateral moraine and the IRD event in Kane Basin is striking and may well be meaningful, but we will also have to consider the wide uncertainties in the dating of both these

events (dR and 14C dating uncertainties in Kane Basin of several hundred years and uncertainties of up to 1.7 ka for the lateral moraine). If we were to consider that unit 3 (the IRD event) is in fact the sole result of calving from the Humboldt Glacier during its retreat and that unit 1 was deposited by the collapse of glacial ice in Kennedy Channel, it would entail that the deglaciation of northern Kane Basin was either (1) relatively fast, (2) that it occurred simultaneously to the opening of Kennedy Channel, or (3) that post-glacial sediments at the core location were eroded by the opening of Kennedy Channel, given that the seismic profile seems to indicate that the core reached till. When taking into account age uncertainties, it is also tempting for us to consider that the change in sediment sources at ∼8.1 cal ka BP (that we interpret as the retreat of Humboldt Glacier in eastern Kane Basin) corresponds to the abandonment of the lateral moraine dated by Reusche et al.

We will definitely include these alternative interpretations in our revised version. As you mention in your review, it is important that we do not over-interpret our dataset without at least giving alternative narratives. Referee #1 also advised us to acknowledge more evidently our uncertainties on the dating of the units we present and the interpretations we propose.

We hope to submit a revised version as soon possible.

Yours sincerely,

Eleanor Georgiadis on behalf of the authors

―――――――――――――――

---

## Author Comment (AC3) · 13 Sep 2018

Dear referee 1,

Thank you again for your review of our work. Please find attached our point-by-point response to you comments relevant to our initial manuscript, and the revised version of this manuscript following you comments and those of referee 2.

Yours sincerely,

On behalf of all co-authors, Eleanor Georgiadis

Please also note the supplement to this comment:
https://www.clim-past-discuss.net/cp-2018-78/cp-2018-78-AC3-supplement.pdf

[Figure]

[Figure]

We would like to thank referee 1 again for allowing us the opportunity to submit a revised version of our work, which we believe has been greatly improved by the detailed review provided by referee 1.

We previously issued a short reply to the referee addressing the main comments that were made in the review. We have taken into account referee 1's suggestions in our revised version which we have submitted, and we would also like to provide a point-by-point response to the original revision in which our responses are in blue font.
* * *
Review by anonymous referee 1 and our responses

Thank you for the opportunity to review Georgiadis and colleagues paper. They present new data, including grain-size, CT, XRF, and radiocarbon, from sediment core AMD14-Kane2b from Kane Basin and discuss implications for the deglaciation of Nares Strait. They infer a major deglacial event, the opening of Nares Strait, from an IRD event and XRF geochemistry. It is a good dataset and is suitable for publication in Climates of the Past. However, there are a few important issues in the discussion and data treatment that should be addressed before this manuscript is accepted for publication. First, I like to praise how the paper focuses on a detailed description of the core stratigraphy on depth. The inclusion of Table 2 in addition to Figure 5 make it very easy for me, the reader, to understand the stratigraphy of the core and the author's interpretation of that stratigraphy. In my view, the most important take away from this paper is the clear description of the stratigraphy and I applaud the authors for that. My biggest issue with the paper is how the authors make statements regarding the meaning of the data and then fit their interpretations to that model. This is particularly true for the XRF data. With the detailed grain-size data set the authors have generated, it would be much more informative to learn about the relationship between sediment geochemistry and grain-size based on Kane Basin data instead of importing conceptual models from vastly different depositional environments. This would make the results of this study much more convincing and help other researchers working in the region. At a minimum, the authors need to be clearer about what is their interpretation and what is supported by data in the results section. I recommend adding a figure showing the relationships between XRF element counts and particle size in the various lithologic units, as this relationship (or lack of relationship) is central to many of the interpretations made by the authors. I would also like to see the authors expand their discussion to include how their data compare to another marine perspective on the Holocene deglaciation of Nares Strait by Jennings et al. (2011). Although the paper is referenced in the introduction and the discussion, the authors do not address why their age for the opening of Nares Strait is younger. I believe the two observations can be reconciled, but it is worth a discussion by the authors as Jennings et al. present faunal and stable isotope data that clearly show the change in oceanographic conditions with the opening of the strait and have a high quality age constraint above the transition at 8,328-8,528 cal yrs BP ($\triangle R$ = 335±85) based on Neogloboquadrina pachyderma sinistral. I consider these to be more reliable evidence than semi-quantitative bulk-sediment geochemistry and an IRD event layer.

We would like to thank you for your encouraging review and overall appreciation of the sedimentological study in our paper. We have tried to address your comments concerning the geochemical study and have also added a comparison of our data with Jennings et al. (2011) and Reusche et al. (2018) in the discussion section of our revised version.

**Fig. 1.** Reply to referee1

[revised manuscript text omitted]

---

## Author Comment (AC4) · 13 Sep 2018

We would like to thank referee 1 again for allowing us the opportunity to submit a revised version of our work, which we believe has been greatly improved by the detailed review provided by referee 1. We previously issued a short reply to the referee addressing the main comments that were made in the review. We have taken into account referee 1's suggestions in our revised version which we have submitted, and we would also like to provide a point-by-point response to the original revision in which our responses are in blue font.
* * *
**Review by anonymous referee 1 and our responses**

Thank you for the opportunity to review Georgiadis and colleagues paper. They present new data, including grain-size, CT, XRF, and radiocarbon, from sediment core AMD14-Kane2b from Kane Basin and discuss implications for the deglaciation of Nares Strait. They infer a major deglacial event, the opening of Nares Strait, from an IRD event and XRF geochemistry. It is a good dataset and is suitable for publication in Climates of the Past. However, there are a few important issues in the discussion and data treatment that should be addressed before this manuscript is accepted for publication. First, I like to praise how the paper focuses on a detailed description of the core stratigraphy on depth. The inclusion of Table 2 in addition to Figure 5 make it very easy for me, the reader, to understand the stratigraphy of the core and the author's interpretation of that stratigraphy. In my view, the most important take away from this paper is the clear description of the stratigraphy and I applaud the authors for that. My biggest issue with the paper is how the authors make statements regarding the meaning of the data and then fit their interpretations to that model. This is particularly true for the XRF data. With the detailed grain-size data set the authors have generated, it would be much more informative to learn about the relationship between sediment geochemistry and grain-size based on Kane Basin data instead of importing conceptual models from vastly different depositional environments. This would make the results of this study much more convincing and help other researchers working in the region. At a minimum, the authors need to be clearer about what is their interpretation and what is supported by data in the results section. I recommend adding a figure showing the relationships between XRF element counts and particle size in the various lithologic units, as this relationship (or lack of relationship) is central to many of the interpretations made by the authors. I would also like to see the authors expand their discussion to include how their data compare to another marine perspective on the Holocene deglaciation of Nares Strait by Jennings et al. (2011). Although the paper is referenced in the introduction and the discussion, the authors do not address why their age for the opening of Nares Strait is younger. I believe the two observations can be reconciled, but it is worth a discussion by the authors as Jennings et al. present faunal and stable isotope data that clearly show the change in oceanographic conditions with the opening of the strait and have a high quality age constraint above the transition at 8,328-8,528 cal yrs BP ($\triangle R = 335 \pm 85$) based on Neogloboquadrina pachyderma sinistral. I consider these to be more reliable evidence than semi-quantitative bulk-sediment geochemistry and an IRD event layer.

We would like to thank you for your encouraging review and overall appreciation of the sedimentological study in our paper. We have tried to address your comments concerning the geochemical study and have also added a comparison of our data with Jennings et al. (2011) and Reusche et al. (2018) in the discussion section of our revised version.

I have included specific comments related to each section and line comments for each section below.

Section/Line comments:

Title: As a suggestion, perhaps include the geographic location of your study site, such as "a marine perspective from Kane Basin," to draw more attention to your article from researchers working in or interested in Kane Basin.
We agree, "from Kane Basin" has been added to the title.

Abstract: The abstract is fine and includes the major interpretations made by the authors. My only general recommendation, beyond the line comments would be to include qualifiers like around before the dates. For example, changing "at 8.3" to "around 8.3" and "from 7.5 to 1.9" to "from about 7.5 to 1.9." There is a great deal of uncertainty in your age model and it is always good to indicate that these are just estimates of the events, especially when you are not reporting the uncertainty associated with those event ages (which would be difficult to quantify with a program like CLAM).
We agree, we have added qualifiers relative to our uncertainties on the exact timing of event.

Page 1 Line 12: For clarity, perhaps rewrite "that translate into ice sheet configuration in the strait" to "that provide new insight to the ice sheet configurations in Nares Strait."
We agree, we have rephrased this sentence as you suggest.

Page 1 Line 14: It is not clear, even after reading the paper, what you mean by "unstable sea surface conditions." It would be helpful to clarify what this means and maybe what data it is based on.
We have specified "unstable *sea ice* conditions and glacial activity" meaning that our record of fluctuating clast content attest to either fluctuating sea ice occurrence or glacial activity such as calving.

Page 1 Line 20: Change sediment source to sediment geochemistry
We agree, this change has been made.

Introduction: This is a nice introduction and clearly introduces the problem you are trying to address.

Page 2 Line 4: I recommend starting a new paragraph after discussing the modern observations and beginning the discussion of the Holocene observations.
We agree, this change has been made.

Page 2 Line 6: I would suggest changing the word admitted. "It is now widely admitted" reads awkward
We agree, we have rephrased this part.

Page 2 Line 8: You could also include the palaeoceanographic evidence from Hall Basin (Jennings et al., 2011).
We agree, we have added this detail and we also mention Reusche's et al. (2018) recent paper in our introduction.

Page 2 Line 19: Include the name of the core
The name of the core has been added.

Page 2 Line 19: Perhaps also indicate that you are presenting radiocarbon data. For example, "sedimentological, geochemical, and geochronological data . . ."
We agree, this has been added.

Regional Settings: This is a nice overview of the regional oceanography and geology. I would just be careful about making the jump from regional geology to geochemical signatures for source regions. While it makes sense that Ca is probably from the carbonate rich regions and the K is from gneiss regions, we don't have a good sense from your paper (or previous work) what the sub-ice bedrock is or what the geochemical variability of these units (like the siliciclastic sedimentary rocks found throughout the region). I would also include a call out to your Figure 2 when you discuss the regional geology.
We have added a section (4.2) where tried to clarify that the spatial variability in modern sediments (in terms of mineralogy) in Kane Basin attests to the delivery of different material depending on the sediment provenance (Kravitz et al., 1976). We have also made mention of a paper that shows a similar distribution pattern of elemental concentrations in modern sediments (Kravitz, 1994) where sediments in eastern Kane Basin are enriched in Fe compared to sediments in western Kane Basin. We thus propose to rather use the Fe/Ca ratio to trace sediment sources in Kane Basin, since Ca is more likely to originate from western or northern Carbonates and Fe likely comes from eastern gneissic material.
We suspect that K is probably a better tracer of gneissic material (e.g. Bervid et al., 2017), but we admit that we do not have any tangible evidence to support this in Kane Basin. We believe that grain size and sediment source may be linked in core AMD14-Kane2b. Both the change to a coarser matrix ca. 280 cm (8.3 cal ka BP) and the decrease in K counts may be associated with the retreat of the GIS in eastern Greenland. As mentioned in the preliminary reply to your review, we hope that future investigations by Caron et al. will provide more insight into this matter. We have tried to better express our uncertainties when using XRF data as a source signal throughout the paper.

Materials and Methods: The CT, grain-size, and XRF methods all seem appropriate and sufficiently described. However, as noted in the line comments, I do not find the conceptual framework for the XRF interpretation convincing and needs further support from data and/or references. The grain-size statement would fit much better in the results section, after a more direct comparison of the 'grain-size sensitive elements' to the detailed grain-size record the authors have generated. This

would be the first direct observation I am aware of for Kane Basin or Nares Strait and would be a helpful observation for others working in the region.

We would like to thank you for bringing this to our attention. We have added a small section on the correlation between the XRF data and grain size in the results (4.2) that we hope is more convincing. We have also added the graphs of K/Ti and K vs. grain size in the supplementary data.

The choice of △ R, 240 years, seems reasonable and is similar to the recent Jakobsson et al. (2018) choice of 268 ± 82 years. I would have liked to see uncertainty included in the calibration of the dates, as the authors acknowledge the pre-bomb △ R range could have been as high as 335 years, but I doubt that would have changed the overall interpretation, as the uncertainty with respect to the material dated in this study is likely larger than the uncertainty associated with the choice of △ R (with 35% of the reported ages rejected in the age model). Given this uncertainty, perhaps it would be better to present the chronology in the results section, after discussing the radiocarbon results and specific justification for which ages you accept/reject.

We discuss our age model in a new result section (4.1) of the revised manuscript.

Page 4 Line 5: Change section number from 2 to 3. Likewise, subheading numbers need to be changes from 3.2 to 3.1 and 3.3 to 3.2

This has been corrected.

Page 4 Line 8: Does u-channel need to be capitalized?

We believe the "u" in U-channel is capitalised (e.g. the British Ocean Sediment Research Facility, BOSRF, uses "U-channel", link in citations).

Page 4 Line 8: Define CASQ acronym

CASQ is the abbreviation of CAlypso SQuare. We hope that the rephrasing of the definition is clearer.

Page 4 Line 11: Define INRS acronym

We have added the definition.

Page 4 Line 27 to Page 5 Line 2: The authors need to do a better job justifying their interpretation of the XRF data. The sentence that begins "The sedimentary rocks from northern Kane Basin and Nares Strait . . ." needs to be supported by data or a reference. I am not convinced that these are the geochemical signatures of each source region otherwise. I also do not think it is convincing based on the low resolution geologic map presented as Figure 2. Additionally, the second sentence that begins "Additionally, heavy elements Ti and Zr reflect the grain size variation . . ." is often true; however, Ti and Zr can also be influenced by provenance changes. The cited references for this statement involve study areas on the Oman Margin, NW Mexican Margins, and Gulf of Cadiz (and a fourth by Correns, from 1954, which is not listed in the reference listed) not Kane Basin. I could list other studies from different regions which show Zr variations are not related to lithic particle size, such as Phillips et al. (2014) in the Bay of Bengal. The point I am making is that I would like to see Georgiadis and colleagues make observations based on the data set they have generated from

Kane Basin, rather than fitting their data to a preconceived notion. I would also argue that the authors are in a great position to do exactly that, as they have generated a detailed grain-size record to go along with their XRF geochemistry.

Again, thank you for bringing this to our attention. As mentioned previously, we have added a section accordingly.

Page 5 Line 7: The supplementary figure could be moved to the main text, as the manuscript only has six figures. They are good data–why bury them in a supplement when you don't need to?

We agree, this change has been made.

Page 5 Line 29: Indicate what settings you used in CLAM 2.2 (i.e. is this a polynomial fit? Or linear interpolation?)

This has been added in both the main text (3.3) and the legend of Fig. 4.

Page 5 Line 31 to Page 6 Line 3: This paragraph would be a better fit for the results section after a discussion of the radiocarbon results and specific criteria for rejected dates.

We agree, this paragraph is now in the results section of the paper (4.1).

Results and Interpretations: As stated earlier, I think the authors do a great job discussing the stratigraphy of the core, the depositional processes that may be important in creation of the deposit, and, for the most part, saving the narrative for the discussion.

Page 6 Line 12: Can you say anything about the thickness of unit 0 given that the acoustic waves are likely attenuated? I think your interpretation is correct, I just don't think you have any evidence for the thickness of the unit.

We agree that we cannot give any information on the thickness of unit 0 and have taken this part out.

Page 6 Line 17: Is the evidence for 'The elemental composition of the sediments is largely dominated by Ca throughout the core' based on the XRF data? If so, even after your normalization, semiquantitative count differences between elements don't necessarily mean weight percent differences. I think you would need a more quantitative method and/or a calibration to make that statement.

Yes, it is based on the XRF data, but this statement was supposed to be merely descriptive (it is what the XRF data shows), not interpretative (we do not mean to say that this reflects the absolute Ca content).

Page 6 Line 18: Why does K reflect source changes and not particle size changes. This needs to be explained further and supported by data and/or references.

We have demonstrated the good correlation between K/Ti and grain size. We now propose to use Fe/Ca as a source tracer.

Page 7 Line 4-5: Perhaps clarify that the larger particles are relative to overlying units, as it seems Unit 1B is finer grained than Units 1A and 1C.

We have added this.

Page 7 Line 27-30: Indicate that these are likely sources or your interpreted sources.
We have tried to clarify throughout the paper that interpretations of the XRF data in terms of sources are to be considered cautiously.

Page 8 Line 5: I would change "A significant portion of these sediments derived from eastern Kane Basin gneisses" to something like "Relatively high K counts likely reflect an increase contribution from eastern Kane Basin gneisses."
We have changed the phrasing in this sentence and in others that shared the same ambiguity.

Page 9 Line 22-25: Like other comments, at this point state the geochemical evidence for your interpretation. The contribution of northern and/or eastern carbonates doesn't increase slightly, Ca counts increase slightly which is likely consistent with an increased contribution of northern and/or eastern Kane Basin carbonates.
We have made this change.

Page 9 Line 27: What is Unit E? Do you mean Unit 5?
Yes, we did mean unit 5.

Page 10 Line 5-6: Make sure that you are discussing the data. There wasn't a change in the delivery of gneiss material. There was a change in K and Ca, which you interpret as a change in provenance. But as you have already discussed, there is also a big change is sediment grain-size.
We have rephrased this sentence. There is indeed a change in grain size toward a coarser matrix which may have led to the K signal being attenuated by other elements. However, the main interpretation, which is the distance to the core site, would likely remain unchanged regardless of the sediment source, since this is also based on the reduced delivery of fine material (by meltwater plumes).

Page 10 Line 15: The authors indicate that two mollusk shells are younger than expected and claim that they were likely remobilized by bioturbation. In one case this would indicate movement of about 80 cm downward in the core (UGAMS-24308). Is this a reasonable interpretation for a mollusk shell? Are there any references that could be used to support that interpretation? Would the simpler interpretation be to accept the younger dates and attribute the older dates to reworking, as you do for the older part of the age model?
We have removed the two younger ages from our dataset. This comes after verification in the cruise report that the mollusc samples which were subjected to those 14C measurements were collected on the ship's deck immediately after removal of the core lid and before cleaning of the 1-2 cm-thick sediment layer in contact with the lid. This sediment layer is suspected to be subjected to important remobilization during coring operation. All other sampling for 14C dating were conducted on clean sediment material within the U-Channels. The excellent correspondence of the three 14C dates measured at ca. 60 cm further support our choice to remove the above mentioned 14C measurements from our original dataset.

Discussion The review of the terrestrial data and comparison to your core data is good. As stated in the intro, I think this paper is missing a comparison to the other important marine perspective–from Hall Basin. It is also my opinion that many of the interpretations are a bit speculative in the discussion section, such as the influence of the 9.3 ka cold event, but I think, because it is in the discussion only, it is okay. However, it detracts from one of the main and well-supported findings of the paper (in the same section), that your region of Kane Basin was deglaciated by 9 ka and that provides an important constraint for your Figure 6. In this instance and others, you might consider focusing your discussion on the major and best supported findings so that they are the main focus of the reader's attention.

Page 11 Line 16: Not necessary for this paper as it was only available online after you submitted your paper to Climates of the Past, but you may also be interested in a recent cosmogenic study by Reusche et al. (2018) for your future work.
Thank you for pointing out this paper to us. We have taken it into account in our revised version since it provides such good discussion material for our study and was also suggested by referee 2.

Page 11 Line 24: Do you mean Figure 5?
We were making reference to Figure 7 which illustrates the deglaciation of the core site ca. 9.0 cal. ka BP and the proximity of the GIS at the beginning of our record.

Page 13 Section 5.3: I do not find the presence of old foraminifera convincing evidence for a Northern Nares Strait sediment source. Obviously, there is something complicated happening with radiocarbon, as 35% of your reported ages are rejected and not used in your age model. Part of your argument may be that these are on mixed benthic forams; however, you have other mixed benthic forams which are clearly too old elsewhere in you core (e.g. SacA-46003) without similar sedimentological indicators. I think this part of the discussion could use a little work and dive a bit more into the uncertainties of your chronology and the constraints from the Hall Basin record (and its uncertainties).
We have made changes to this part of the discussion (5.3) where we have confronted our interpretations with Jennings et al. (2011). Interpretations concerning the old foraminifera have been removed.

Page 13 Line 32 and Page 14 Line 4: You use the term 'gneiss signal.' As I've been critical of in other parts of this review, you present no data that K is a proxy for gneiss or sediment sourced to a specific region.
We have made the change.

Conclusions:
Page 15 Line 17: Change section number from 5 to 6.
We have corrected this.

Page 16 Line 2: Authors claim this is the first non-land view point of the deglaciation of Nares Strait. This is not true, as a northern perspective of the events was documented by Jennings et al. (2011)

using a marine sediment core from Hall Basin. The authors need to change this to a southern perspective, or a Kane Basin perspective.
This has been corrected, we have changes "Nares Strait" to "Kane Basin".

5  Figures:

Figure 3: In the caption, indicate that the age model is a polynomial, spline, or linear interpolation (whatever you used) and indicate if the shading is the one sigma or two sigma uncertainty of that model fit.
10  This information has been added.

Figure 5: In the caption, indicate that the plotted radiocarbon dates are only the ones you accepted to use in your age-depth model and make a call out to Table 1, so it is clear to the reader that there are other radiocarbon data.
15  This has been changed.

References Cited in Review:

Jakobsson, M., Hogan, K.A., Mayer, L.A., Mix, A., Jennings, A., Stoner, J., Eriksson, B., Jerram, K.,
20  Mohammad, R., Pearce, C., Reilly, B., Stranne, C., 2018. The Holocene retreat dynamics and stability of Petermann Glacier in northwest Greenland. Nat. Commun. 9, 2104. https://doi.org/10.1038/s41467-018-04573-2

Jennings, A.E., Sheldon, C., Cronin, T.M., Francus, P., Stoner, J., Andrews, J., 2011. The Holocene
25  history of Nares Strait: Transition from glacial bay to Arctic-Atlantic throughflow. Oceanography 24, 18–33.

Phillips, S.C., Johnson, J.E., Giosan, L., Rose, K., 2014. Monsoon-influenced variation in productivity and lithogenic sediment flux since 110 ka in the offshore Mahanadi Basin, northern Bay
30  of Bengal. Mar. Pet. Geol., Geologic implications of gas hydrates in the offshore of India: Results of the National Gas Hydrate Program Expedition 01 58, 502–525. https://doi.org/10.1016/j.marpetgeo.2014.05.007

Reusche, M.M., Marcott, S.A., Ceperley, E.G., Barth, A.M., Brook, E.J., Mix, A.C., Caffee, M.W.,
35  2018. Early to Late Holocene Surface Exposure Ages From Two Marine-Terminating Outlet Glaciers in Northwest Greenland. Geophys. Res. Lett. 0. https://doi.org/10.1029/2018GL078266

References cited in our reply:

40  http://www.boscorf.org/sites/default/files/documents/Uchannel%20sampling%20procedure%20final.pdf

[revised manuscript text omitted]
 weighted average age for the retreat of the Humboldt Glacier was presented in Reusche et al., (2018) as 8.3 ±1.7 ka BP, but two samples were suspected of contamination by previous exposure. When these two samples are omitted, the weighted average age of the retreat becomes 8.1 ±0.6 ka BP, which reconciles our dating of the retreat of the Humboldt Glacier with that evidenced by Reusche et al. (2018). However, given the uncertainties in both our radiocarbon dataset (analytical errors and ΔR uncertainties) and in their surface exposure dataset, it is difficult to distinguish whether the retreat of the Humboldt glacier was near-coeval with 
[revised manuscript text omitted]

---

## Author Comment (AC5) · 13 Sep 2018

We would like to renew our many thanks to referee 2 for the pertinent review of our paper and for allowing us the opportunity to submit a revised version of our work. The main comments have been addressed in a short reply, but we also provide a point-by-point response to referee 2 here, along with a revised version of our paper. In this revised version, we propose alternative interpretations of our record as we compare it to Jennings et al. (2011) and Reusche et al. (2018). We however favour our original scenario, which is that unit 3 represents the opening of Kennedy Channel ca. 8.3 cal. ka BP. Our responses are in blue font.
* * *
**Review by anonymous referee 2 and our responses**

I have read this manuscript with great interest. It uses data collected from a single, strategically located core from Kane Basin to infer changing paleoenvironmental conditions in response to ice sheet recession in Nares Strait from early Holocene to present.

The paper is clearly written and well organized. The main data include lithofacies descriptions from visual core descriptions and CT scanning, CT number as a measure of density, grain size data to infer sedimentation processes, and XRF data to infer sediment provenance. The core chronology is based on 13 radiocarbon dates on benthic foraminifera and molluscs. A local reservoir correction of 240 years was applied to calibrate the dates.

Figure 4, the 3.5 kHz chirp profile over the core site shows that the core penetrates the section essentially to acoustic basement which is interpreted to be subglacial till at the site, meaning that the core should contain a sequence that includes initial deglaciation, opening of Nares Strait, and continued ice recession from the ocean all along the strait.

I think that the interpretation of the events in Kane2B is reasonable, but there is an alternative interpretation of the sequence of events that also should be considered. In part this stems from a new article that includes new information about early Holocene moraine ages of the Humboldt Glacier: Early to Late Holocene surface exposure ages from two marine terminating outlet glaciers in northwest Greenland Melissa M. Reusche, Shaun A. Marcott, Elizabeth G. Ceperley, Aaron M. Barth, Edward J. Brook, Alan C. Mix and Marc W. Caffee Accepted manuscript online: 13 JUL 2018 12:00AM EST | DOI: 10.1029/2018GL078266

In this paper surface exposure ages indicate that an early Holocene lateral moraine of the Humboldt Glacier was abandoned by 8.3 ±1.7 ka. Could retreat from this stable position be the source of the increased IRD and other features of unit 3A,B,C rather than being a signal of the opening of the strait?

Thank you for bringing this paper to our attention. This is indeed a possibility. We have tried to cover this in our revised version (sections 5.1 and 5.2).

However, even if we were to switch to a preferred interpretation of our record as: unit 1C being the opening of the strait at 9.0 cal. ka BP and unit 3 being deposited by the retreat of the Humboldt Glacier, this would still not reconcile the discrepancy in the dating of this event in our core vs. that

of Jennings et al. (2011). In Jennings et al. (2011) the age of the IRD-rich unit is estimated at 8.6 cal. ka BP with a $\Delta R=240$, bringing it closer in age to unit 3 than to unit 1C (and the $\Delta R$ is likely to be higher in Hall Basin than in Kane Basin so it is likely to be even closer to unit 3).

5   Alternatively, the retreat from the moraine might signal the ice recession that lead to opening of the strait, which would support the argument made in this paper. A connection between retreat from the moraine and opening of the strait was not made by Reusche and others, mainly because in the literature the opening of Nares Strait is suggested to be earlier by some hundreds of years.
We do favour this scenario. As we show in our revised version (sections 5.2 and 5.5), the retreat of
10   the Humboldt Glacier evidenced by Reusche et al. (2018) may even be coeval to the retreat of the ice margin from the core site ca. 8.1 cal. ka BP. It is possible that major instabilities in GIS/IIS began ca. 8.3 cal. ka BP, leading to the collapse of glacial ice in Kennedy Channel, but that the GIS was then stabilised by colder conditions during the "8.2 event". After this, ice retreat could have resumed leading to the abandonment of the lateral moraine in Washington Land and the retreat of glacial ice
15   in eastern Kane Basin ca. 8.1 cal. ka BP.

An alternate scenario for the stratigraphy of the core is: Unit 1, ice proximal, reflects opening of Nares Strait and Unit 3 represents retreat of the Humboldt Glacier. Given the uncertainties of the reservoir corrections used all along Nares Strait and through the Holocene, I think it would be useful
20   for the authors to consider this alternative scenario.

If the benthic DR in Kane Basin is larger than the 240 years used, then the deeper unit 5 could represent opening of the Strait, which would mean that all of Kane 2B was deposited after the strait opened.
25   We did not consider that the $\Delta R$ in Kane Basin may have been larger than 240 year and that unit 1C may correspond to the opening of Kennedy Channel AND to the retreat of the Humboldt glacier ca. 8.3 cal. ka BP (Reusche et al., 2018). It would indeed mean that, similarly to the alternative explanation now provided in the paper (unit 1C is the opening of the strait), all of the AMD14-Kane2b core was deposited after the opening of the strait. Since the outcome of this interpretation
30   concerning core AMD14-Kane2b would be the same as with the main alternative, and that it's a bit more of a stretch to link the abandonment of the moraine with our unit 1C rather than unit 3, we choose not to explore this other alternative so as not to further complicate our discussion section.

Can other arguments about environmental changes one might expect from the opening of the strait
35   be brought to bear to support one interpretation or the other? It is clear that the age presented for the opening of Nares Strait by Jennings et al. (2011) is somewhat older than 8.3 ka, even when the higher DR of 335 years is used. In any case, I think it is reasonable to consider an alternative interpretation of the events. In the end, one can favor one interpretation over the other based on the evidence. But I don't think the origin of Unit 3 can be proven with the data shown, especially given
40   the uncertainties in the reservoir corrections and the short time span involved. However, it would be really interesting if the moraine recession, Unit 3, and the opening of Nares Strait were linked.
We have tried to cover the main alternative scenarios in our revised version, while favouring our original scenario. It is true that one cannot be absolutely certain of which scenario is accurate based on the evidence presented in this paper alone, although we remain convinced that by confronting

our record with Jennings et al. (2011) and Reusche et al. (2018) our initial narrative is more robust. We believe that future work on core AMD14-Kane2b will provide more tangible evidence for our proposed scenario.

5   Minor comments:

P1. Line 8. 8 ka 14C calibrates to 8.7 cal with DR=240 I think. I think the other calibrations of dates from the literature are done correctly with the reservoir correction added back before the dates were calibrated. But it is worth checking. Especially with regard to Figure 6. I missed a description of how
10  the dates were recalibrated. And there might need to be a list of the dates shown on the maps with the 14C age and the cal age along with the references etc.
We have corrected 8.5 to 8.2 cal ka BP (which is 8 ka 14C with ΔR=240 ±51). So as to give a very general idea of the history of Nares Strait, we focus, in this part of the introduction, on the early work carried out in Nares Strait and the dating of the period during which it was blocked by glacial ice,
15  which was very approximate at the time.
We have added a table in the supplementary information with the mollusc radiocarbon data published in previous studies and our calibrations.

P2. Line 18. What about St. Onge and St. Onge, 2014?
20  This reference has been added.

P3, 6. punctually. . . not sure what you mean.
The current velocity in Robeson Channel was measured instantaneously on board the cruise ship in Münchow et al. (2007). This very high speed may thus not be representative of the yearly average
25  speed but we are unaware of any long-term current measurements in Robeson Channel.

P3, 7,8. Make this 2 sentences.
We have rephrased this sentence.

30  P3, 15. Ice arches only block the sea ice. . . and allow the liquid freshwater through.
The idea that ice arches not only block sea ice but also control water export through the strait is based on Münchow's (2016) study that compares fluxes through Nares Strait between 2003 and 2006, when the ice bridge was present 5-8 months/year, and 2006 and 2009, when the ice arch fail to form or was only present 2 months. The study finds that "volume flux increased by 45%, ocean
35  freshwater flux increased by 69%, and ice freshwater flux increased by 46% from the first to the second period".

P3, 17. change 'associated to' to 'outlets of'?
This change has been made.
40
P4, 16, 28. Calculate and use the ± on the DR.
This has been done (ΔR=240 ±51 years).

P4, 31. Change 'drastic' to 'major'?
This has been changed.

P6, 7. You may be overinterpreting these thin units. Change 4 to 6 cm. They all appear to be ice proximal and there is variability in the sedimentation along an ice margin that can be reflected in the sediments you have.
We have tried to simplify the interpretations of unit 1 by omitting interpretations on the small-scale variations within this unit.

P6, 18. Change 'oscillates' to 'varies'?
This has been changed.

P6, 23. Evolve, decrease
This has been corrected.

P6, 24. Meltwater plumes and iceberg rafting
This has been corrected.

P8, 20. And P13, 6. What species were dated? Can these be a mixture of >50 k and contemporaneous forams? It seems unlikely that forams would look fresh after being entrained in a current. This is not very convincing about forams coming from Hall Basin.
As with molluscs, foraminifera species were not formerly identified in this study. The sample is likely to have included mainly *Cassidulina reniforme* and *Islandiella norcrossi* based on the species present in other horizons of unit 3B.

P9, 1. Along with
This has been corrected.

P9, 11. Suggest you delete 'discretely'.
This change has been made.

P9, 21. Svendsen
This has been corrected.

P9, 25. Keep the format you were using with the unit name and details at the start of the line.
This has been rephrased.

P9, 27. Unit5 rather than E.
This has been corrected.

P10, 3. Suggest you skip the double negative and say . . . 'We rule out hypothesis 3 . . .
This has been rephrased.

P10 in general. It would be helpful to name the species dated...both mollusk and foram when possible. There are some species to avoid and it is useful to have the info.
Thank you, we will definitely not make this mistake again.

P11, 6,7. I don't think the second part of this sentence is right. Suggest you end sentence at 'strait'. And delete the rest except keep the reference.
This part has been removed from the sentence.

P12, top paragraph. I think this part is over interpreted. And there is quite a bit of >800 micron material. Basically this unit just seems ice proximal and does not necessarily suggest the sequence of environments.
We have tried to simplify the discussion of unit 1 by omitting interpretations on the small-scale variations within this unit.

P12. Discussion of Fig. 6. How about putting on a through h designations on your panels. Then you can refer to the right panel in your discussion.
This has been done. Thank you, it does make it easier to follow.

P12, 20. Change 8.3 to 8.5. 8.5 is what is on the map.
This has been corrected.

P12, 32. Coarse fraction content is not only influenced by sea ice. I think this is overinterpreted.
P13, 32. Sea ice and ice bergs certainly are not mutually exclusive. I think you are over interpreting sea ice with your grain size data because you have a lot of coarse material in your units. Icebergs carry all grain sizes.
We have tried to modulate our interpretation of grain size with regards to sea ice. We however find it interesting to compare the Agassiz atmospheric temperature record and sea ice conditions in nearby locations with potential sea surface conditions (or glacial activity) in Kane Basin.

P14, 5. Outset change to onset?
This change has been made.

P14, 8. A stretch for you to interpret sea ice here with just the grain size.
We have tried to modulate our interpretation.

P16, first para. Your interpretation is not unique...there is at least one other alternative. It is best to explore that. The DR is poorly constrained.
We have added the alternative scenarios in the discussion section of our revised paper (sections 5.1, 5.3 and 6).

Fig. 3. Add tick marks to the axes. Add the lithofacies units to the depth scale.
These have both been added.

Fig. 4. Can you remove the interpretation and put it in a panel below? One cannot see the boundaries on the data.

The interpretations have been removed from the seismic profile.

5  Figure 5. why is lithofacies 1 2 boundary listed as 9.1 when an underlying age is 9?

This was a mistake; it has been corrected to 9.0 cal. ka BP.

Fig. 6. Explain how you calibrated the ages in the map and list the ages in a table with their pertinent information.

10  This has been added. A table and map of the mollusc ages have been added in the supplementary data.

Table 2. fix the tick marks and add the date levels where they coincide with the examples of lithofacies

15  This has been fixed.

**References cited in our reply:**

[revised manuscript text omitted]
 weighted average age for the retreat of the Humboldt Glacier was presented in Reusche et al., (2018) as 8.3 ±1.7 ka BP, but two samples were suspected of contamination by previous exposure. When these two samples are omitted, the weighted average age of the retreat becomes 8.1 ±0.6 ka BP, which reconciles our dating of the retreat of the Humboldt Glacier with that evidenced by Reusche et al. (2018). However, given the uncertainties in both our radiocarbon dataset (analytical errors and ΔR uncertainties) and in their surface exposure dataset, it is difficult to distinguish whether the retreat of the Humboldt glacier was near-coeval with 
[revised manuscript text omitted]

---

## Author Response (AR1)

**Editor Decision: Publish subject to minor revisions (review by editor)** (30 Oct 2018) by Ed Brook

Comments to the Author:

Please accept my sincere apologies for the delay in handling your paper. I hope it was not too much of an inconvenience.

Thank you for having provided us with the final corrections for our manuscript. We hope that you will be satisfied with our responses and the changes we made.

The editor's comments are in black font; our responses are in blue font. Changes made to the revised manuscript in response to the editor's comments are highlighted in yellow.

I feel that your revisions have for the most part satisfied the reviewers concerns and that the new manuscript is appropriately cautious. There are just a few points I would ask you to address in a second revision:

Editor comments on revision of cp-2018-78

Page 1, line 22, "geochemistry" is misspelled.

Thank you, we have corrected this, along with the other misspelling of this word Page 7.

Section 3.3 has the same title as section 4.1. Please make a change in one of them.

Thank you, we have changed the titles as followed:

**3.3 Chronology and radiocarbon dating in Nares Strait **

**4.1  Age model and sedimentation rates in core AMD14-Kane2b**

Page 6, line 26. It would probably be best to translate A contrario to English as the phrase (sadly) is not common for English speakers (use "in contrast").

Thank you, *"A contrario"* has been changed to "In contrast".

Page 6. Chronology section. Unless I missed it there is no specific discussion of the rejection of the outliers in the age model. This should be discussed in the text.

We tried to address the rejection of outliers in this paragraph (Page 7 line 13-16). We have added a sentence before this specifying that we removed these 4 radiocarbon ages from the age model to make this clearer:

"Fourteen of the $^{14}$C yielded consistent ages along a smooth spline, while four outlying radiocarbon ages were excluded from the age model. Only one mollusc fragment was dated (at 301.5 cm) and yielded an age of >43 ka and is thus clearly remobilised (Table 1). A mollusc shell at 238.5 cm yielded a radiocarbon age about 1 ka older than expected and is the only specimen we suspect to be affected by the *Portlandia* effect. Two mixed benthic foraminifera samples yielded ages older than expected and most likely include older specimens."

Page 18, lines 10-20. I think this section should be simplified. All of the ages are within uncertainty of each other regardless of the outliers, so it seems awkward to reject outliers then claim better

agreement. Instead you can simply describe the level of agreement and that you can't reject the hypothesis that the moraine abandonment and GIS retreat were synchronous. Given the story line you wish to present this may take a little reorganization of the text.

Given the doubt expressed by Reusche et al. (2018) concerning the possible natural contamination of two samples in their dataset, we think it is a shame not to recalculate the age of the abandonment of the moraine, especially since it coincides so well with the likely retreat of Humboldt Glacier dated ca. 8.1 cal Ka BP in our record. If the editor however thinks that it is best to simply the discussion, we propose these changes to the paragraph:

"The timing of the retreat of the GIS in eastern Kane Basin corresponds remarkably well with the aforementioned abandonment of a lateral moraine by the Humboldt Glacier (Reusche et al., 2018). ~~The weighted average age for the retreat of the Humboldt Glacier was presented in Reusche et al., (2018) as 8.3 ±1.7 ka BP, but two samples were suspected of contamination by previous exposure. When these two samples are omitted, the weighted average age of the retreat becomes 8.1 ±0.6 ka BP, which reconciles our dating of the retreat of the Humboldt Glacier with that evidenced by Reusche et al. (2018). Howeverboththeirthe retreat of the Humboldt glacier was near-coeval with~~ this event was linked to the deglaciation of Kennedy Channel at ~8.3 cal. ka BP, or whether it was delayed until ~8.1 cal. ka BP, after the cold "8.2 event" that may have brought a short period of stability to the GIS."

Page 19, line 21. The word "accurate" seems awkward to me here.

We have made the change to "true", hoping it is better suited.

The table in the supplementary information that contains published radiocarbon ages does not have a title or caption.

Thank you, we have added a title for the table and changed the numbering of the supplementary information accordingly.

One of the reviewers made this comment:

P8, 20. And P13, 6. What species were dated? Can these be a mixture of >50 k and contemporaneous forams? It seems unlikely that forams would look fresh after being entrained in a 20 current. This is not very convincing about forams coming from Hall Basin.

Your response is below, but as far as I can tell no changes were made to the text for this. Could you address this issue further?

As with molluscs, foraminifera species were not formerly identified in this study. The sample is likely to have included mainly Cassidulina reniforme and Islandiella norcrossi based on the species present in other horizons of unit 3B.

We believe we discuss the possibility that older specimens were included in the sample, and then provide the alternative being that the forams may have been transported from Hall Basin as a secondary explanation in the Results and Interpretation section 4.3 of our paper: (Page 7 line 17-22):

"Concerning the old age yielded from the mixed benthic foraminifera picked in this subunit, the age model shows that these foraminifera were remobilised. It is possible that a small quantity of pre-Holocene foraminifera was mixed in with living fauna. This would imply that sediments pre-dating the last glaciation (>22 cal. ka BP) were preserved under the extended GIS and IIS in Nares Strait, and were eroded and transported to the core site during the deposition of subunit 3B. An alternative explanation is that the sample is composed of postglacial specimens of a similar age which were eroded from the seabed and transported to the site."

In the discussion section of the manuscript, we proposed in our revised version not to discuss the older age in this unit as a result of referee 2's comment, leaving the reader the choice of our two interpretations mentioned in section 4.3. We originally thought that the older foram age in unit 3B which coincided with the age of post-glacial sediment dated in Hall Basin was part of the evidence showing that this unit represented the opening of Kennedy Channel. We deleted this paragraph so as not to build our preferred story on this radiocarbon age, as suggested by referee 2.

This is the deleted paragraph in the discussion section 5.3 (Page 16 line 11-17) in response to referee 2's comment:

[revised manuscript text omitted]